# Conformation and dynamics of soluble repetitive domain elucidates the initial β-sheet formation of spider silk

Nur Alia Oktaviani[1], Akimasa Matsugami[2], Ali D. Malay[1], Fumiaki Hayashi[2], David L. Kaplan [ID] [3] & Keiji Numata [ID] [1]

The β-sheet is the key structure underlying the excellent mechanical properties of spider silk. However, the comprehensive mechanism underlying β-sheet formation from soluble silk proteins during the transition into insoluble stable fibers has not been elucidated. Notably, the assembly of repetitive domains that dominate the length of the protein chains and structural features within the spun fibers has not been clarified. Here we determine the conformation and dynamics of the soluble precursor of the repetitive domain of spider silk using solution-state NMR, far-UV circular dichroism and vibrational circular dichroism. The soluble repetitive domain contains two major populations: ~65% random coil and ~24% polyproline type II helix (PPII helix). The PPII helix conformation in the glycine-rich region is proposed as a soluble prefibrillar region that subsequently undergoes intramolecular interactions. These findings unravel the mechanism underlying the initial step of β-sheet formation, which is an extremely rapid process during spider silk assembly.

---

[1] Biomacromolecules Research Team, RIKEN Center for Sustainable Resource Science, 529 5F Main Research Building, 2-1 Hirosawa, Wako, Saitama 351-0198, Japan. [2] Advanced NMR Application and Platform Team, NMR Research and Collaboration Group, NMR Science and Development Division, RIKEN SPring-8 Center, Yokohama, 1-7-22 Suehiro-cho, Tsurumi-ku, Kanagawa 230-0045, Japan. [3] Department of Biomedical Engineering, Tufts University, 4 Colby Street, Medford, MA 02155, USA. Correspondence and requests for materials should be addressed to K.N. (email: keiji.numata@riken.jp)

Spider silk has attracted great interest because of its superior mechanical properties and potential for industrial and biomedical applications[1]. Spider dragline silks consist of high-molecular-weight proteins (250–350 kDa) that contain conserved and relatively small N-terminal domains (NTDs) and C-terminal domains (CTDs) (16.5 and 10.5 kDa, respectively), as well as a long repetitive domain[2,3]. The molecular structure of silk fibers has been characterized using solid-state NMR and X-ray diffraction[4–7]. Consensus motifs identified in the repetitive domain are mainly composed of polyalanine stretches (4–12 alanine residues), which form β-sheets in a crystalline region, and the glycine-rich region (GGX), which exhibits $3_1$-helix structures in an amorphous region[4–6,8,9].

Prior to being spun into insoluble fibers, the precursors of dragline silk are stored as a highly concentrated protein solution in the major ampullate gland. Several studies have shown that the NTD and CTD display strong pH dependence in terms of conformation and are essential for controlling the pH-dependent assembly of silk proteins[10–13]. The presence of the highly conserved CTD of dragline silk is also important for directing fiber formation in vivo[14].

Although, the conserved NTD and CTD play important roles in spider silk protein self-assembly, the conformational details and role of the long repetitive domain in the soluble form are less well defined and are the focus of the present study for several important reasons. First, several studies have reported that in the absence of the CTD and NTD, a long repetitive domain of spidroin is still able to form fibers[15–17], and number of the repetitive domains are proportional to the strength of the recombinant silk fibers[15,16]. Second, although several studies have indicated that the CTD and NTD are important for maintaining the solubility of spider silk in the major ampullate gland, the size of the CTD and NTD is much smaller than that of the repetitive domain. Therefore, the high solubility of spider silk in the ampullate gland and the fiber-forming processes likely do not depend solely on the CTD and NTD. Furthermore, the dominating role of the repetitive domain in terms of the size and fiber functions suggests that additional insights are needed in these regions of silk proteins. Based on these points, we hypothesized that the conformation of the soluble repetitive domain plays a significant role in fiber formation and determines the high solubility of spider silk. Another important point is that the conformation of the soluble precursor of spidroin (spider silk protein), particularly for the repetitive domain, remains unclear. Vibrational circular dichroism (VCD) spectroscopy data revealed the presence of PPII helix and random coil structures in soluble dragline silks[18]; whereas, NMR data of $^{13}C$-labeled spidroin from spider glands indicated that these proteins only consist of random coil conformations, which are highly dynamic[9,19,20]. However, the mechanism underlying the structural transition from disordered conformations to stable spider silk fibers with a β-sheet structure remains unclear because this transformation might lead to nonspecific aggregation. Therefore, we address a fundamental and specific question: is the repetitive domain fully a random coil or does it possess a certain regularity in structure in the soluble form that might explain the mechanism of transformation of soluble spider silk to insoluble silk fibers?

Although, the NTD and CTD display pH-dependent conformations, the pH-gradient effect in the gland on the repetitive domain of spider silk proteins remains unknown. The importance of the pH effect on the repetitive domain was highlighted in a recent report, which demonstrated the importance of dityrosine formation in the repetitive domain of silkworm silk to promote β-sheet formation[21]. The formation of dityrosine in the repetitive domain was catalyzed by the presence of peroxidase[21], in which the activity of peroxidase is optimal at a slightly acidic pH

(pH 6.0–6.5)[22]. Thus, investigations of the pH effect on the conformation of the repetitive domain might be useful for identifying the comprehensive mechanism of β-sheet formation of spider silk, which until now has remained poorly understood.

In this study, we investigated the molecular conformation and dynamics of different numbers of repetitive domains of dragline silk at neutral pH using solution-state NMR, far-UV circular dichroism (CD) and VCD spectroscopies. Our results demonstrate that the repetitive domain of dragline silk consists of two major populations: random coil (~65%) and PPII helix (~24%). We propose that the PPII helix population in the glycine-rich region might serve as the prefibrillar form of spider dragline silk and may contribute to the efficiency of the spinning process because the PPII helix can easily undergo intramolecular interactions in response to shear forces and dehydration by forming a reverse turn. Additionally, the limited flexibility of the glycine-rich region might function to maintain the solubility of spider silk in the ampullate gland. pH titration studies of the repetitive domain at acidic pH values (from 7 down to 2.6 to emulate events in the gland in vivo) show no conformational change, suggesting that the pH changes in the spinning gland does not affect this prefibrillar form. These findings provide new insights into the initiation step of rapid β-sheet formation in spider silk proteins.

## Results

**Solubility of the repetitive domain of dragline spider silk.** Five different lengths of the recombinant repetitive domain (i.e., monomer, dimer, trimer, hexamer, and 15-mer) were expressed using *Escherichia coli* BL21(DE3). The repetitive domain of spidroins were designed based on the amino acid sequence of major ampullate spidroin 1 (MaSp1) from *Nephila clavipes* (GenBank accession ID M37137.2) (Fig. 1a). Protein purity was evaluated by sodium dodecyl sulfate polyacrylamide gel electrophoresis (SDS-PAGE) (Supplementary Fig. 1). The monomer, dimer, trimer, hexamer, and 15-mer could be concentrated to 2.2 mM (for 15-mer, concentration ~100 g L$^{-1}$) without any detectable precipitation. Additionally, pH titration (from 2.6 to 11.3) of the 15-mer did not cause observable precipitation.

**Overall structural propensity of the repetitive domains.** The overall structural propensities of the repetitive domain of spider silk at low concentrations (15–35 µM) were determined via CD spectroscopy. The monomer spectra had a minimum at ~200 nm, which is typical of an intrinsically disordered protein[23,24]. In contrast, the spectra of the dimer, trimer, hexamer, and 15-mer demonstrated minima at ~196 nm; in particular, the CD spectra of the hexamer and 15-mer displayed small but detectable maxima at ~215 nm (Fig. 1b), which is consistent with the PPII helix conformation[23,24]. Consistently, the CD spectra of multiple repetitive domains (dimer, trimer, hexamer, and 15-mer) detected minima similar to those of the spectra of regenerated spider silk proteins from *Nephila edulis* in fresh conditions (incubation time $t = 0$)[25].

The presence of the PPII helix population of the repetitive domains was evaluated in $D_2O$ at a high concentration (8 mM) via VCD spectroscopy in the amide I region corresponding to the carbonyl stretch. The VCD spectrum of the 15-mer at pH 7 and 25 °C in the amide I region demonstrated a maximum at 1662 cm$^{-1}$ and an observable minimum at 1636 cm$^{-1}$ (Fig. 1c), suggesting the presence of the PPII helix[18,26]. In contrast, the delta absorbance of the VCD spectrum of the monomer decreased significantly, indicating that the monomer predominantly contains the unordered random coil population rather than the PPII helix population[27]. Thus, the presence of the PPII helix population in the 15-mer at neutral pH was confirmed via CD

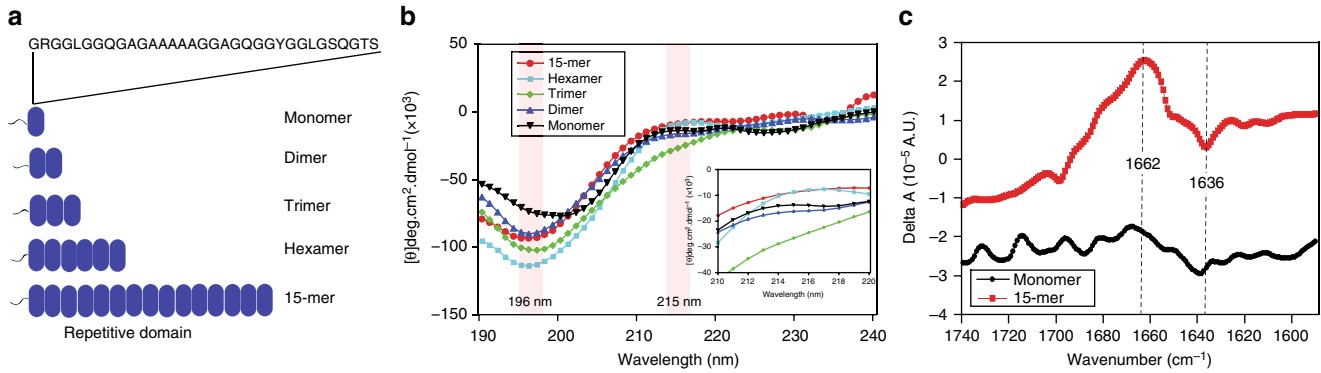

**Fig. 1** Repetitive domains (Hexamer and 15-mer) contain PPII helix conformation. **a** Repetitive domain of spidroin (i.e., monomer, dimer, trimer, hexamer, and 15-mer) used in this study. The repeated amino acid sequence is indicated. **b** CD spectra of the different repeat constructs at 10 °C. The monomer (black) exhibited a minimum at ~200 nm; whereas, the dimer (blue), trimer (green), hexamer (cyan), and 15-mer (red) displayed minima at ~196 nm (indicated in shaded region). The hexamer and 15-mer demonstrated slight maxima at ~215 nm (indicated in the shaded region and insert). The CD data are presented as smoothed averaged of 10 measurements (**c**) VCD spectra of the 15-mer and monomer at 25 °C and pH 7 in D$_2$O. The 15-mer showed a distinctive maximum at 1662 cm$^{-1}$ and minimum at 1636 cm$^{-1}$, which is typical of the PPII helix population[18,26]. In contrast, the overall delta absorbance of the monomer was significantly lower than that of the 15-mer, suggesting a higher proportion of the random coil conformation

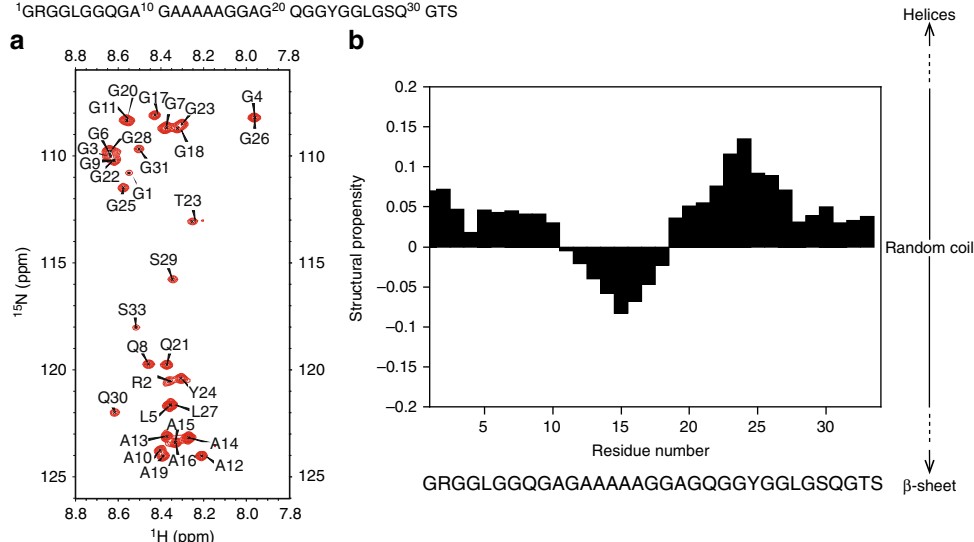

**Fig. 2** Conformational differences between the polyalanine and glycine-rich regions. **a** 2D $^1$H–$^{15}$N HSQC spectrum of the 15-mer at pH 7 and 10 °C. **b** Structural propensity of the 15-mer at pH 7 and 10 °C

and VCD. This finding is consistent with a previous study reporting the presence of the PPII helix population of intact native spider dragline silk using VCD spectroscopy[18].

**Detailed structural propensities of the repetitive domains**. The concentrations of the ($^{13}$C, $^{15}$N) repetitive domains of the spidroins used for the NMR measurements were ~1.5–2.2 mM (monomer: 11.7–17.0 g L$^{-1}$, dimer: 15.8–23.0 g L$^{-1}$, trimer: 19.6–28.8 g L$^{-1}$, hexamer: 31.4–46.0 g L$^{-1}$, 15-mer: 69.3–100 g L$^{-1}$). As shown in Fig. 2a, the 2D $^1$H–$^{15}$N heteronuclear single quantum coherence (HSQC) of the 15-mer displayed poor chemical shift dispersion (<1 ppm) of the amide proton domain, which is a characteristic of an unfolded protein spectrum.

The 2D $^1$H–$^{15}$N HSQC spectra of the monomer, dimer, trimer, and hexamer also demonstrated poor chemical shift dispersion of the amide proton signals, suggesting that these proteins were also unfolded under conditions similar to those of the 15-mer (Supplementary Fig. 2B). The presence of His-tag and cloning site (Thr-Ser) did not affect the conformation of the repetitive domain (Supplementary Fig. 2, Supplementary Discussion).

All backbone (Cα, Hα, C, N$^H$, and Cβ) and ~84% of the total side chain resonances of the repetitive domains at pH 7 and 10 °C could be assigned using 2D and 3D NMR measurements. When the backbone chemical shifts of the repetitive domain were translated into structural propensity values using the neighbor-corrected structural propensity calculator (ncSPC)[28], all repetitive domains were predominantly unfolded as indicated by a low structural propensity value (<0.15). When the number of repetitive domains was increased, the polyalanine regions (AGAAAAAGGAG) displayed a small but observable increase in β-sheet propensity as indicated by the structural propensity values of the trimer, hexamer, and 15-mer (closer to −0.1).

Observable low β-sheet propensities occurred in the polyalanine sequence. The glycine-rich regions in every construct of the repetitive domain (monomer, dimer, trimer, hexamer, and 15-mer) exhibited low-helical propensities (Supplementary Fig. 3). A small increase in helical propensity was also found for the first three residues of the monomer repeat unit. This increase likely occurred because the initial repeat residues were located proximal to the His-tag containing the DDDDKAMAAS sequence, which

showed a higher helical propensity (Supplementary Fig. 4). Furthermore, the high resolution of the 2D H(N)CO spectra demonstrated that in the dimer, trimer, hexamer, and 15-mer, the backbone carbonyl chemical shifts of [26]GLGSQGTS[29] and [1]GR[2] residues were shifted relative to those of the monomer (Supplementary Fig. 5). The carbonyl chemical shift differences between the 15-mer and monomer and between the 15-mer and dimer were plotted as a function of the residue number (Supplementary Fig. 6A and 6B). Since the backbone chemical shift of the intrinsically disordered protein is only influenced by direct neighbor residues ($i-1$ and $i+1$), chemical shift differences of more distant residues (>2 residues from the interface of repetitive domain) between the 15-mer and monomer indicated that the 15-mer was not a fully random coil protein (Supplementary Fig 6A)[30]. In addition, the chemical shift difference between the 15-mer and dimer in the end terminal was similar to the chemical shift difference between the 15-mer and monomer, while the chemical shift difference between the 15-mer and dimer in the middle region was close to zero (Supplementary Fig 6B). This finding suggests that the dimer is not fully random coil, similar to 15-mer. These chemical shift changes also indicated a slight difference in structural propensity in that region (Supplementary Fig. 3).

**Backbone dynamics of the repetitive domains**. The backbone dynamics of the monomer and 15-mer were probed with the $^{15}NT_1$, $^{15}NT_2$ relaxation and $\{^1H\}$–$^{15}N$ heteronuclear nuclear Overhauser effect (NOE) under two different magnetic fields (700 and 800 MHz) (Supplementary Fig. 7). The shorter $^{15}NT_1$ and longer $^{15}NT_2$ relaxation times of the monomer suggested that the monomer had a shorter rotational correlation time and faster dynamics than the 15-mer[31]. Consistently, the 15-mer had higher NOE values at both 700 and 800 MHz NMR than the monomer, suggesting that the 15-mer was less flexible than the monomer (Supplementary Fig. 7C and 7F). Furthermore, $\{^1H\}$–$^{15}N$ heteronuclear data showed higher NOE values of the 15-mer in the glycine-rich region than in the polyalanine region (Fig. 3a).

Spectral densities of the 15-mer and monomer were calculated at J(0), J(ωN), and J(0.87 ωH) based on $^{15}NT_1$, $^{15}NT_2$ $\{^1H\}$–$^{15}N$ heteronuclear NOE data for the monomer and 15-mer using 2 different magnetic fields, 700 and 800 MHz, which corresponded to 0, 70, 80, 609, and 696 MHz (Supplementary Fig. 8). The most striking results were obtained for the spectral densities of the 15-mer and monomer at a frequency of 0 MHz (J(0 MHz)) (Supplementary Fig. 8C) because this spectral density reflects the rotational correlation times ($\tau_c$) of those molecules. The J(0) values of 15-mer (~5.5 ns) suggested that the rotational correlation time of the 15-mer was longer than the rotational correlation time of the monomer, which presented a J(0) of ~2.2 ns. This finding also indicated that the 15-mer underwent slower motion than the monomer. Taken together, all the dynamics data indicated that the 15-mer did not behave as a fully random coil protein because it rotated more slowly than intrinsically disordered proteins, which normally have a very short spectral density J(0) (0.5–2.2 ns)[32] Additionally, higher rotational correlation time of the 15-mer might be also attributed to transient interaction between the repetitive domains since PPII helix conformation often shows intermolecular hydrogen bonds[29].

**Secondary structure population of the repetitive domain**. The distribution of the secondary structure population of the 15-mer was determined using δ2D[33]. As depicted in Fig. 3b, the 15-mer is composed of two major populations, random coils (65%) and PPII helix (24%). The larger population of the PPII helix (~25% of the total secondary structure population) was attributed to the

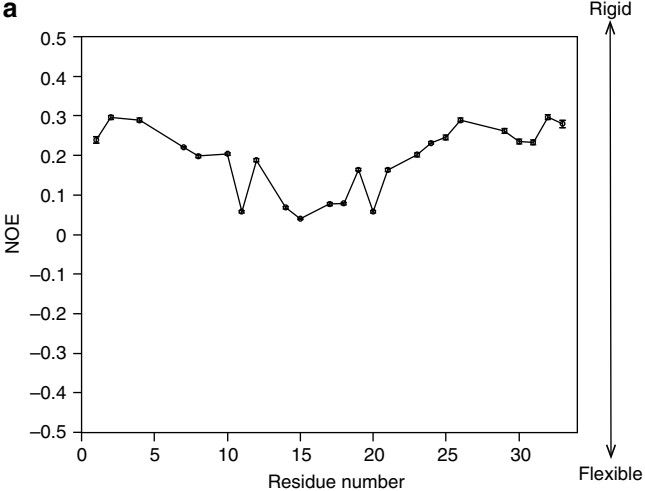

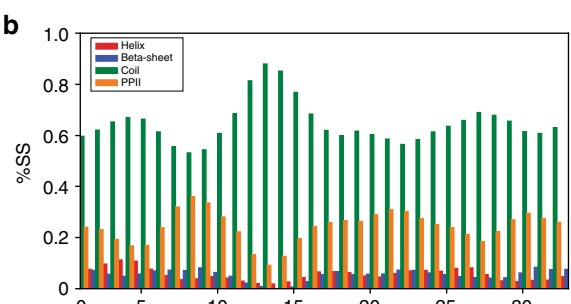

GRGGLGGQGAGAAAAAGGAGQGGYGGLGSQGTS

**Fig. 3** Backbone dynamics and secondary structure population of repetitive domains. **a** $\{^1H\}$–$^{15}N$ heteronuclear NOE of the 15-mer. **b** Secondary structure population of the 15-mer based on backbone chemical shifts. Overlapping peaks in $\{^1H\}$–$^{15}N$ heteronuclear NOE spectra were not included in the $\{^1H\}$–$^{15}N$ heteronuclear NOE graph of the 15-mer. %SS denotes % secondary structure. Error bars in $\{^1H\}$–$^{15}N$ heteronuclear NOE graph was calculated based on base plane noise in the $^1H$–$^{15}N$ HSQC spectra with and without proton saturation (equation 3)

glycine-rich region. In the polyalanine region, the PPII helix population decreased and the random coil population increased. These data were also corroborated by the $^3J$ HNHα coupling constant data of the repetitive domain (Supplementary Fig. 9). In the 15-mer, the average $^3J$ HNHα coupling constant of the polyalanine region was 5.6 ± 0.5 Hz, which is closer to the typical $^3J$ HNHα coupling constant of the random coil alanine (5.8 Hz)[34]; whereas, the average $^3J$ HNHα coupling constant of the glycine-rich region of the 15-mer was 6.5 ± 0.5 Hz, which is somewhat closer to the average $^3J$ HNHα coupling constant for a peptide containing the PPII helix (6.50 Hz)[35]. Specifically, we found that the average $^3J$ HNHA coupling constant of Gln of the 15-mer at 10 °C was ~6.6 ± 0.2 Hz, which is similar to the previously reported $^3J$ HNHA coupling constant of the model PPII helix peptide at 10 °C (6.8 Hz)[35]. As comparison, the average $^3J$ HNHα coupling constant of the glycine-rich region of the monomer was 6.4 ± 0.7 Hz, and the $^3J$ HNHA coupling constant of Gln of the monomer was 7.3, 6.9, and 5.9 Hz for Gln-8, Gln-21, and Gln-30, respectively, demonstrating that the actual $^3J$ HNHA coupling constants of Gln of the monomer were closer to the coupling constant value of Gln that is normally found in a random coil peptide (7.1 Hz)[35].

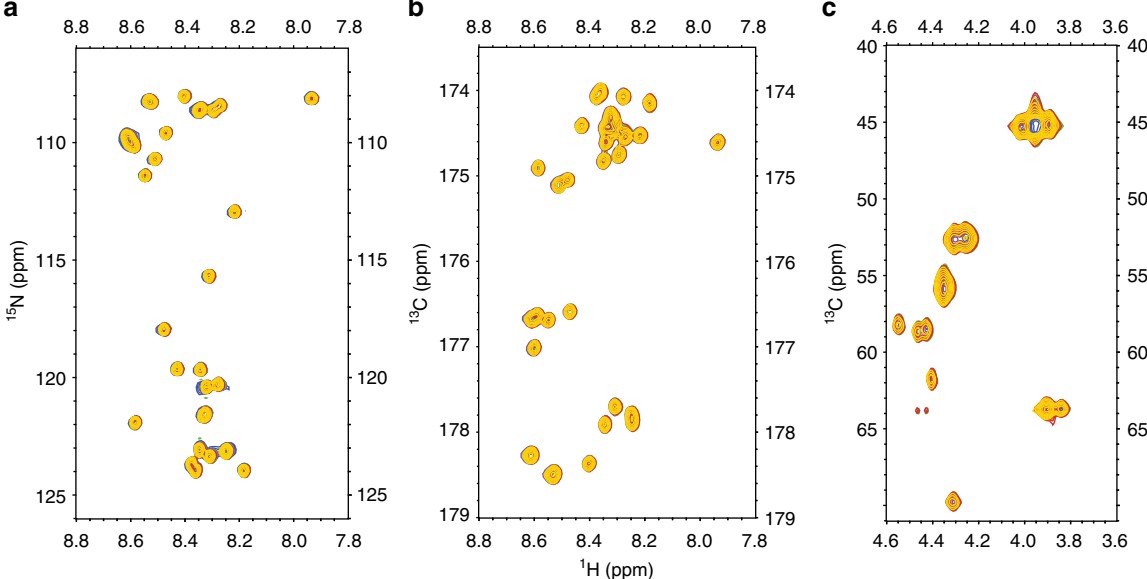

**Fig. 4** No conformational changes observed in the repetitive domain at an acidic pH. Neither chemical shift changes nor line broadening effects were observed in the repetitive domain (15-mer) in the pH range from 7 to 2.6 at 10 °C. **a** Overlay of the 2D $^1$H–$^{15}$N HSQC spectra of the 15-mer. **b** Overlay of the 2D projection H(N)CO of the 15-mer, which provides the correlation with $H^N_{(i)}$–C′$_{(i-1)}$. **c** Overlay of the 2D $^1$H–$^{13}$C HSQC spectra in the aliphatic region. Blue: pH 7; purple: pH 6; green: pH 5; red: pH 4; and yellow: pH 2.6

**Effect of pH on the repetitive domain**. The CTD and NTD are known to display a strong pH dependence for their conformations in vitro[10–12,36], and these results are similar to the findings for the pH gradient in the spider gland[37]. To understand the pH effect on the structure of the repetitive domain at an acidic pH, several 2D NMR experiments, namely, $^1$H–$^{15}$N HSQC, $^1$H–$^{13}$C HSQC and 2D projection of H(N)CO, were performed to follow the backbone (Cα, Hα, N$^H$, and C′) chemical shifts of the 15-mer from pH 2.6 to 7.0. We found that neither chemical shift changes nor line broadening occurred in that pH range (Fig. 4). This result implies that the repetitive domains of MaSp1 do not undergo any conformational changes in response to a decrease in pH.

The pH effect on the repetitive domain at a basic pH was probed using 2D CON and 2D $^1$H–$^{13}$C HSQC aliphatic experiments. After further pH titration under basic conditions, only marginal chemical shift changes and structural propensities (Supplementary Fig. 10A, 10B and Supplementary Fig. 11) were observed around Tyr residues at pH 10.2. This result can be explained by deprotonation of the Tyr side chain. The tyrosine side chain p$K_a$ was determined using $^{13}$Cε as a chemical shift reporter (Supplementary Fig. 10C). Our pH titration data (pH 7.0–11.3) indicated that the p$K_a$ of the Tyr side chain of the 15-mer was 10.25 ± 0.02 (Supplementary Fig. 10D), which is identical to the typical value determined for solvent-exposed Tyr[38]. Although the Tyr side chain p$K_a$ of the repetitive domain was determined in the absence of salts (NaCl, KCl), the presence of high concentrations of salt (~0.4 M) only affected amino acid p$K_a$ values ranging from ~0.2–0.9 pH units as previously reported[39].

**Concentration effect on the repetitive domain**. The intermolecular interactions of the repetitive domain were also evaluated in this study by monitoring the backbone chemical shift (N$^H$, H$^N$, Cα, Hα) of the 15-mer at different concentrations. Based on our NMR data, we found no change in the backbone chemical shifts of the repetitive domain at different concentrations (concentration range: 23.1 g L$^{-1}$ (0.5 mM) up to 46.2 g L$^{-1}$ (1 mM)) (Supplementary Fig. 12A, 12B and 12C). This finding

indicated that there was no change in the population of soluble repetitive domains, as indicated by the CD and VCD spectra (Fig. 1b, c). In addition, the ratio between the signal-to-noise (S/N) values of the signals from the 2D $^1$H–$^{15}$N HSQC spectrum of the 15-mer at 1 mM (4 scans) and 0.5 mM (16 scans) was approximately one and uniform for the entire sequence. This result suggested that observable changes did not occur in the interaction or dynamics of the repetitive domain at higher concentrations (Supplementary Fig. 12C). This finding also suggested that no entanglement effect was observed for the repetitive domain under this condition. In the presence of mechanical and/or chemical stimuli (e.g., shear forces, kosmotropic ions, etc.), the entanglement effect of the repetitive domain might be affected at higher concentrations.

**Temperature effect on the repetitive domain**. The effect of temperature on the conformation and dynamics of the soluble repetitive domain were investigated using NMR and CD. As the temperature increased from 10 to 25 °C, the signal intensities of the $^1$H–$^{15}$N HSQC spectra decreased because of the rapid exchange of amide protons to water (Supplementary Fig. 13A). However, based on the overlay of the Cα-Hα region from the $^1$H–$^{13}$C HSQC spectra, no chemical shift changes were observed from 10 to 25 °C (Supplementary Fig. 13B), suggesting that there were no significant changes in the conformation of the soluble repetitive domain. At a higher temperature (15 °C), the {$^1$H}–$^{15}$N heteronuclear NOE data for the repetitive domain consistently showed that the glycine-rich region had more limited flexibility than the polyalanine region (Supplementary Fig. 13C). The CD spectra of the 15-mer at different temperatures (Supplementary Fig. 13D) also demonstrated minima at 196 nm and weak maxima at ~215 nm, which persisted from 10 to 30 °C, suggesting the presence of the PPII helix population[23]. However, the decrease in the ellipticity of the 15-mer at ~215 nm at higher temperatures (Supplementary Fig. 13D) indicated that the PPII helix population of the repetitive domain decreased as the temperature increased[40].

## Discussion

In this study, we investigated the conformations and dynamics of the soluble repetitive domains of *N. clavipes* in the absence of the terminal domains. Although, the mechanical properties of spider dragline silk are diminished in the absence of the terminal domains[16,41], previous studies reported that individual domains of spider silk function independently, and no stable interactions were observed between these domains[42,43]. Therefore, we consider that our study on conformations and dynamics of repetitive domains in the absence of terminal domains might be highly relevant to the conformations and dynamics of the repetitive domains when considered in the presence of terminal domains.

The structural propensity of the polyalanine region of the repetitive domain is similar to that of the random coil; although, the negative structural propensity score in this region suggests that a β-sheet precursor already occurred in the soluble form. A similar finding was reported for the soluble β-sheet precursor of the monomer and dimer of Aβ40[44]. Intriguingly, the soluble β-sheet precursor in the polyalanine region of spidroin is similar to the β-sheet region of spider silk fiber as demonstrated by solid-state NMR data[4–6,45], but the conformation in this region is different from the one found in the β-sheet region (GAGAGS) in the silkworm silk of *Bombyx mori*, which forms a repeated β-turn type II structure prior to spinning[46].

Our finding suggest that PPII helix conformation in the glycine-rich region is retained before and after the spinning process of spider dragline silk[4–6,9]. This finding is also consistent with a previous study demonstrating that the PPII helix conformation also carried –GGX– and –GPG– motifs in *Argiope trifasciata* flagelliform silk[47]. The PPII helix, which is also recognized as a $3_1$ helix or polyglycine II[48,49], is markedly more flexible than the α-helix and β-sheets[50], although this conformation is still less dynamic than the random coil conformation. Therefore, the presence of the PPII helix in the glycine-rich region contributes to more limited flexibility of the 15-mer than of the monomer. The presence of the PPII helix population in the glycine-rich region is reflected by more limited flexibility in this region than in the polyalanine region, even at a temperature of 15 °C. The PPII helix conformation is important because the geometry of this conformation allows the protein chain to progress directly to form a reverse turn[51].

Previously, solid-state NMR indicated that the conserved LG (G/S)QG motif in the glycine-rich region (Supplementary Note 1) adopts a turn conformation in MaSp1 fibers[52]. Such findings are supported by our results, which showed that the structural propensity (Supplementary Fig. 3) and backbone carbonyl chemical shifts in the GLGSQGTS region differ from those of the monomer and other repetitive domains (Supplementary Fig. 5 and 6). These findings suggest that the chemical environment of this motif in multiple repetitive domains with at least 2 repeat units (dimer, trimer, hexamer, and 15-mer), which are able to undergo intramolecular interactions, differs from that in the monomer. Our data also demonstrate that when the number of repetitive domains is increased, the helical propensity of the GLGSQGTS region also increases, suggesting that the PPII helix population is stabilized by the greater number of the repetitive domains (Supplementary Fig. 3). In short repeat lengths, such as the dimer and trimer, the PPII helix population is present but less stable. Therefore, the CD spectra of the dimer and trimer did not display detectable positive maxima at 215 nm, which is a clear indication of a PPII helix population. Furthermore, PPII helix stabilization with a larger number of repetitive domains enables spidroin to readily form intramolecular interactions and explains why stronger spider silk fibers can be achieved when the silk protein contains high numbers of repetitive domains[15].

The propensity of PPII helix conformation in native spider dragline silk had been reported using VCD spectroscopy[18], which is consistent with our far-UV CD and VCD data. This previous study using VCD spectroscopy provided the overall conformation, while the conformation and dynamics of the repetitive domain as a function of amino acid sequence remained unclear. In this study, using solution state NMR spectroscopy, we demonstrated the local conformations and dynamics of the repetitive domain as the function of amino acid sequence. Our data clearly demonstrated that the PPII helix population was distributed over the glycine-rich region; whereas, the random coil population was distributed over the polyalanine region. This finding is consistent with the dynamics data, namely, the glycine-rich region has more limited flexibility compared with the polyalanine region.

Despite the PPII helix population of the 15-mer represents only 25% of the total secondary structure population in the glycine-rich region, the presence of PPII helix populations in the soluble repetitive domains plays an important role in the spinning efficiency because this conformation readily undergoes intramolecular interactions. Here we propose that soluble spider silk occurs in equilibrium with two major populations (Schematic illustration in Fig. 5): random coils (~65%) and PPII helix (~24%). The soluble precursor of the repetitive domain containing the PPII helix serves as the prefibrillar form of the spider dragline silk. The PPII helix conformation is favorable for protein–protein interactions, such as protein complex assembly[50]. This soluble prefibrillar form of spider dragline silk might form irreversible β-sheet structures via PPII helix interactions in the glycine-rich region, which could potentially lead to the formation of silk fibers in response to a change in the biochemical environment, extensional flow, dehydration, and shear forces. The PPII helix interaction in the glycine-rich region is possibly mediated by hydrogen bonds between alpha proton and carbonyl oxygen, similar to the triple helix structure of collagen[29]. In the case of Gln in the glycine-rich region, PPII helix interactions are possibly mediated by hydrogen bonds between Gln side chain and carbonyl oxygen of the neighboring residue[53]. The PPII helix conformations in the glycine-rich regions between repeat domains enables the ultra-rapid conversion into aligned protein polymers during the silk spinning process, as well as pre-ordering of the soluble spidroins, which helps prevent premature aggregation, even at extremely high concentrations.

Interestingly, conformation and dynamics of the recombinant repetitive domain of *N. clavipes* in this study are different from those of the recombinant repetitive domain of *Euprosthenops australis* in the presence of terminal domains[42]. These differences are possibly due to the length of the polyalanine region. The long polyalanine region (14–15 alanine residues) of the repetitive domain of *E. australis* shows helical structure, while the glycine-rich region has random coil conformation. As shown in the current study, a shorter polyalanine region (5 alanine residues) in *N. clavipes* repetitive domain leads to random coil conformation. Our results are in agreement with previous study, which revealed that the conformation of polyalanine region (6–8 alanine residues) of native *Lactrodectus hesperus* dragline silk was random coil[54]. In contrast, the longer polyalanine region in *E. australis* repetitive domain leads to the formation of helical structure[55]. The helical structure of the polyalanine in *E. australis* is also similar to the helical structure of a long polyalanine region (10–14 alanine residues) in *Samia cynthia ricini*[56]. In the case of *E. australis* spidroins, hydrophobicity of the repetitive domain and hydrophilicity of the C-terminal domain cause to form micelle-like structure[42]. Together, these studies suggest that the soluble form of shorter polyalanine region (4–8 alanine residues) of the repetitive domain tends to form random coil, while longer

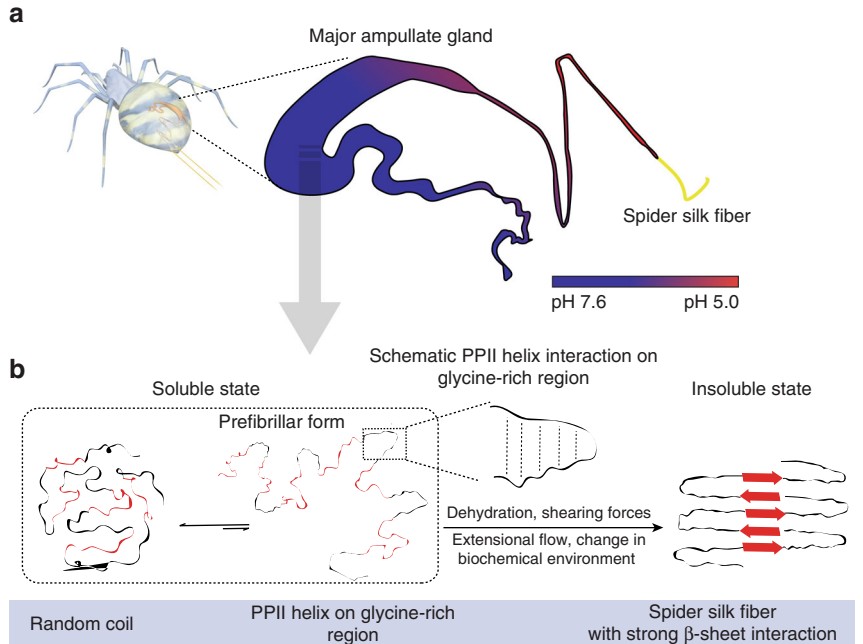

**Fig. 5** Proposed mechanism of β-sheet formation of spider dragline silk. **a** Spider major ampullate gland displays a strong pH gradient (reproduced with permission[70], Copyright 2017, American Chemical Society). **b** Two major structural populations of spider dragline silk proteins, random coil, and PPII helix in the glycine-rich region, occur in soluble form. The presence of the PPII helix in the glycine-rich region is proposed as a soluble prefibrillar form of spider dragline silk. The polyalanine region is shown in red, and the glycine-rich region is shown in black. Schematic illustration of PPII helix interaction in the glycine-rich region is shown in the box. In response to dehydration, shearing forces, extensional flow and changes in the biochemical environment, this prefibrillar form will generate spider dragline fibers through strong β-sheet interactions

polyalanine regions (>10 alanine residues) tend to form helical structure. Furthermore, the recombinant repetitive domain of *E. australis* is easily aggregated even at low concentration (1 g L$^{-1}$)[55]; whereas, the recombinant *N. clavipes* repetitive domain is soluble at relatively high concentration (100 g L$^{-1}$). The difference in solubility of the repetitive domains can be explained based on the dynamic behavior of the glycine-rich region. The limited flexibility of the glycine-rich region of the repetitive domain from *N. clavipes* seems to be essential for maintaining high solubility of spider silk protein by preventing premature aggregation. On the other hand, the high flexibility of the glycine-rich region of *E. australis* repetitive domain leads this domain to be easily aggregated[55].

Our investigation of the acidic pH effect revealed no conformational changes in the repetitive domains, suggesting that pH changes in the spinning gland do not influence the prefibrillar region of the repetitive domain. The conformational change in a protein as a function of pH is normally accompanied by protonation and deprotonation of acid and basic groups. Since no acidic residue is available in the repetitive domain, no conformational changes were observed at acidic pH. However, in vitro studies of native and recombinant spider dragline silk composed of NTD, repetitive domains and CTD demonstrated β-sheet formation at an acidic pH[54,57,58]. Thus, based on the results of the present and previous studies, we concluded that CTD and NTD might be necessary for inducing β-sheet formation in spider silk during the early stage of the spinning process. Furthermore, dehydration, shear forces, and changes in biochemical environment are also involved in the β-sheet formation of spider silk proteins.

Previously, volumetric, and spectrophotometric titration studies of native spidroin (*Nephila edulis*) demonstrated two apparent p*K* values: 6.74 and 9.21[59]. The first p*K* value was hypothesized to originate from the deprotonation of different residues; whereas, the latter p*K* value was thought to be related to

the deprotonation of Tyr side chain residues[59]. In addition, deprotonation of the Tyr side chain has been thought to be involved in the solubility of spider dragline silk at pH values >8.5[60]. In this study, since the p$K_a$ value of the Tyr side chain of the repetitive domain is ~10.3, this indicates that deprotonation of the Tyr side chain was not essential for increasing the solubility of spider dragline silk because the p$K_a$ value of the Tyr side chain at pH 10.3 is too basic for the ampullate gland.

In conclusion, the soluble repetitive domains of spider dragline silk at neutral pH have two major populations: PPII helix (~24%) and random coils (~65%). The PPII helix conformation is more prevalent in the glycine-rich regions, which present less flexibility than the polyalanine region, which contains more random coils. The glycine-rich region with the PPII helix population is proposed as the soluble prefibrillar form, which readily supports the transformation into insoluble silk fiber. Our study also demonstrated that the molecular structure of the repetitive domain is not affected by pH, indicating that the prefibrillar form of the repetitive domain is not influenced by pH. This study provides a better understanding of the initial mechanism of spider silk protein self-assembly and the high solubility of spidroin in the major ampullate gland. Future designs to produce strong artificial spider silk should consider a combination of a large number of repetitive and non-repetitive domains (CTD and NTD) of spidroin as well as a combination of soft (glycine-rich region) and hard (polyalanine region) segments in the repetitive domains to form strong and desirable intramolecular and intermolecular interactions.

## Methods
**Cloning**. The repetitive domain of spidroin was designed based on the consensus repeat (GRGGLGGQGAGAAAAAGGAGQGGYGGLGSQG) derived from the native sequences of the dragline MaSp1 sequences from *N. clavipes* (Accession no. P19837). The monomer, dimer, trimer, hexamer, and 15-mer genes, which correspond to 1, 2, 3, 6, and 15 contiguous copies of this repeat, respectively, were

cloned into the pET30a vector. This vector was modified with a linker containing the restriction sites *Nhe*I and *Spe*I based on previously published papers[61,62]. The repetitive domain genes of each construct were confirmed using DNA sequencing.

**Sample preparation.** The monomer, dimer, trimer, hexamer, and 15-mer were overexpressed using *E. coli* BL21(DE3) cells. A pre-culture was grown at 37 °C and 160 r.p.m. for 16 h by cultivating *E. coli* containing the pET30a-repetitive domain genes in LB medium. Next, the pre-culture was transferred to the main culture. After IPTG (final concentration of 1 mM) induction, the main culture was grown at 30 °C and 160 r.p.m. for 4 h. To purify the repetitive domain of spidroin, every 1 g cell pellet was resuspended in 5 mL of buffer containing 10 mM Tris, pH 7, 8 M urea, and 100 mM $NaH_2PO_4$. The suspension was stirred at room temperature for ~1 h. Next, the suspension was centrifuged at 9000×$g$ for 30 min. The supernatant was collected and applied to a His-trap column (GE Healthcare, Uppsala, Sweden), which was equilibrated using 20 mL of buffer A (20 mM phosphate buffer, 8 M urea, 500 mM NaCl, and 0.5 mM imidazole, pH 7). After the suspension was applied, the His-trap column was washed using 50 mL of buffer A to remove non-specifically bound molecules. A gradient of imidazole was used to elute the repetitive domain using buffer B (20 mM phosphate buffer, 8 M urea, and 500 mM NaCl and 500 mM imidazole, pH 7), and most of the repetitive domain was eluted at a concentration of ~250 mM imidazole. The repetitive domain was concentrated to <100 μL and then purified using a Superdex 200 10/300 GL (GE Healthcare) column in 20 mM phosphate buffer pH 7 with 150 mM NaCl. Salt was removed either by overnight dialysis or by applying the sample to a portable desalting column (PD-10, GE Healthcare, Buckinghamshire, UK). The purities and sizes of the monomer, dimer, trimer and hexamer were checked by 4–20% or 10–20% SDS-PAGE (Supplementary Figure 1A), and the purity of the 15-mer was checked by 4–15% SDS-PAGE (Supplementary Figure 1B).

**CD spectroscopy.** CD measurements were performed using low concentrations (15–35 μM) of unlabeled monomer, dimer, trimer, hexamer, and 15-mer in 10 mM phosphate buffer pH 7 at a temperature of 10 °C. The CD spectra were recorded from 240 to 190 nm in a 1-mm path length quartz cuvette using a JASCO J-820 spectropolarimeter. The wavelength step for the CD measurements was 1 nm, and the scan rate was 500 nm/min. The spectra shown were subtracted from the background and averaged over 10 consecutive scans.

**VCD spectroscopy.** The VCD spectra of unlabeled 15-mer (8 mM or 369.6 g/L) in $D_2O$ at neutral pH were measured at room temperature. As a negative control, the VCD spectra of unlabeled monomer (2 mM) in $D_2O$ at a neutral pH were also recorded at room temperature. The VCD spectra of the 15-mer and monomer had absorbance maxima in the IR spectra ranging from ~0.3 to 0.35. The VCD spectra were measured using a JASCO FVS-6000 system. Baseline corrections were performed by subtracting the spectra of $D_2O$. The VCD measurements were conducted in the region of 2000–850 $cm^{-1}$ at a resolution 4 $cm^{-1}$ and based on an accumulation of 31,992 scans.

**Protein NMR samples.** Each oligomer of the repetitive domain (monomer, dimer, trimer, hexamer, and 15-mer) was overexpressed in *E. coli* BL21(DE3). The repetitive domain was uniformly labeled ($^{13}C$, $^{15}N$) using M9 minimal medium containing (2 g/L) $^{13}C$-glucose and (1 g/L) $^{15}N$-ammonium chloride, supplemented with 30 μg/mL kanamycin. The pre-culture was grown in 100 mL of M9 minimal medium at 30 °C and 160 r.p.m. overnight. This culture was then transferred to the main culture (total volume 1 L) and continuously grown at 37 °C and 160 r.p.m. until it reached $OD_{600}$ ~1, which was followed by 1 mM IPTG. After 4 h of induction, the cells were collected. Doubly ($^{13}C$, $^{15}N$) labeled repetitive domains of spidroin were purified in the same way as purification of unlabeled repetitive domain. The unlabeled monomer, which consisted of 30 amino acids and did not have a His-tag and Thr-Ser cloning site (RGGLGGGQGA GAAAAAGGAG QGGYGGLGSQG), was synthesized by the Research Resources Centre of the RIKEN Brain Science Institute. The NMR samples of repetitive domains were concentrated until the final concentration was ~1.5–2.2 mM in 10 mM phosphate buffer pH 7, which was supplemented with 10% $D_2O$ and 0.1 mM DSS.

**Backbone and side-chain chemical shift assignment.** A series of 2D and 3D NMR experiments, including 2D $^1H$–$^{15}N$ HSQC, 2D $^1H$–$^{13}C$ HSQC (aliphatic region, with the carrier position at 35 ppm), 2D $^1H$–$^{13}C$ HSQC (aromatic region, with the carrier position at 125 ppm), 3D HNCO, 3D HN(CA)CO, 3D HNCACB, 3D (H)C(CO)NH TOCSY and 3D H(CCO)NH TOCSY experiments, was performed using either a triple resonance probe ($^1H$, $^{13}C$, and $^{15}N$) and a JEOL ECA (600 MHz) or a TCI cryogenic probe and a *z*-axis gradient coil of a Bruker spectrometer (700 MHz) at 283 K (10 °C) for unambiguous assignment of the backbone and side-chain chemical shifts of the repetitive domains. All spectra were processed using NMRPipe[63] and analyzed using SPARKY[64]. DSS was used as a reference standard for all NMR signals according to IUPAC recommendations[65]. To investigate the influence of the His-tag and Thr-Ser (TS) cloning site on the chemical shift of the repetitive domains, a 2D $^1H$–$^{15}N$ HMQC experiment using 1024 direct complex points, 128 indirect complex points and 256 scans of 2 mM unlabeled synthetic monomer was performed. The resulting spectrum was compared

with the 2D $^1H$–$^{15}N$ So-fast HMQC spectrum of ~1.5 mM ($^{13}C$, $^{15}N$) monomer containing the His-tag.

**Structural propensities of the repetitive domains.** The structural propensities of the monomer, dimer, trimer, hexamer, and 15-mer were calculated using the ncSPC with all assigned backbone (Cα, Hα, C', Cβ, and $N^H$) chemical shifts[66].

**Backbone dynamics of the monomer and 15-mer.** Backbone amide $^{15}N$ transverse ($T_2$) relaxation experiments using ~1.5 mM ($^{13}C$, $^{15}N$) monomer and ~2 mM ($^{13}C$, $^{15}N$) 15-mer were performed at 10 °C on a 700 and 800 MHz Bruker spectrometer equipped with a TCI cryogenic probe and a *z*-axis gradient coil. The pulse sequence for $^1H$-detected $^{15}NT_2$ relaxation recordings led to a series of 2D $^1H$–$^{15}N$ spectra that were correlated with different $^{15}NT_2$ relaxation delays[67]. The $^{15}NT_2$ relaxation delays were 17, 34, 51, 85, 102, 136, 170, and 204 ms. Each $^1H$–$^{15}N$ correlation spectrum was acquired using 16 scans, a 1 s recycle delay, 1024 direct complex points, and 256 indirect complex points. The acquisition times were 45.7 and 70.3 ms for the direct ($^1H$) and indirect ($^{15}N$) domains, respectively. All data were processed using NMRPipe[63] and analyzed using SPARKY[64].

Backbone $^{15}NT_2$ values were determined by fitting the peak intensities using a single-exponential decay:

$$I(t) = I_0 \exp(-t/T_2), \tag{1}$$

where $I(t)$ is the intensity after a delay time $t$ and $I_0$ is the intensity at time $t = 0$.

Backbone amide $^{15}N$ longitudinal ($T_1$) relaxation experiments using ~2 mM ($^{13}C$, $^{15}N$) 15-mer were performed at 10 °C on a 700 and 800 MHz Bruker spectrometer equipped with a TCI cryogenic probe and a *z*-axis gradient coil. The pulse sequence for $^1H$-detected $^{15}NT_2$ relaxation recordings led to a series of 2D $^1H$–$^{15}N$ spectra correlated with different $^{15}NT_2$ relaxation delays[67]. The $^{15}NT_2$ relaxation delays were 10, 70, 150, 250, 370, 530, 750, and 1150 ms. Each $^1H$–$^{15}N$ correlation spectrum was acquired using 16 scans, a 1 s recycle delay, 1024 direct complex points, and 256 indirect complex points. The acquisition times were 45.7 and 70.3 ms for the direct ($^1H$) and indirect ($^{15}N$) domains, respectively. All data were processed using NMRPipe[63] and analyzed using SPARKY[64].

Backbone $^{15}NT_1$ values were determined by fitting the peak intensities using a single-exponential decay:

$$I(t) = I_0 \exp(-t/T_1). \tag{2}$$

{$^1H$}-$^{15}N$ steady-state NOE values were obtained by measuring two spectra: an initial spectrum recorded without the initial proton saturation and a second spectrum recorded with the initial proton saturation (3 s). {$^1H$}–$^{15}N$ steady-state NOE values of the monomer and 15-mer were determined at 700 and 800 MHz. The steady-state NOE values were determined from the ratios of the average intensities of the peaks with and without proton saturation, and the standard deviation of the NOE values was determined from the background noise level using the following formula[67]:

$$\sigma NOE/NOE = \left( (\sigma I_{\text{sat}}/I_{\text{sat}})^2 + (\sigma I_{\text{unsat}}/I_{\text{unsat}})^2 \right)^{1/2}, \tag{3}$$

where $I_{\text{sat}}$ and $I_{\text{unsat}}$ are the measured intensities of the resonance in the presence and absence of proton saturation, respectively. The noise of the background regions of the spectra, which were recorded with the initial proton saturation and without initial proton saturation, is shown as $\sigma I_{\text{sat}}$ and $\sigma I_{\text{unsat}}$, respectively.

The amide $^{15}N$ relaxation was analyzed using a reduced spectral density mapping approach[68]. This method assumes that the spectral density is relatively constant in the high-frequency region near $J(\omega H)$; therefore, $J(\omega H + \omega N)J(\omega H - \omega N) \sim J(0.87 \omega H)$. The spectral densities at $\omega = \omega N$, $\omega = 0.87 \omega H$ can be obtained from the following formulas:

$$J(0.87 \omega H) = J(\omega H) = \frac{4}{5d^2} \left( \frac{\gamma_N}{\gamma_H} \right) \left( \frac{NOE - 1}{T_1} \right), \tag{4}$$

$$J(\omega N) = \frac{\left[ \frac{1}{T_1} - \left( \frac{7d^2}{4} \right) J(\omega H) \right]}{\frac{3d^2}{4} + c^2}. \tag{5}$$

The spectral densities of $J(0)$ from the two spectrometer frequencies were calculated from the following expression:

$$J(0) = \frac{1}{\beta}\left[\left\{\frac{1}{T_2^{800}} - \frac{\kappa}{T_2^{700}}\right\} - \frac{3d^2}{8}\left\{J(\omega_N^{800}) - \kappa J(\omega_N^{700})\right\}\right.$$
$$\left. - \frac{c_{800}^2}{2}\left\{J(\omega_N^{800}) - \kappa J(\omega_N^{700})\right\}\right.$$
$$\left. - \frac{13d^2}{8}\left\{J(\omega_H^{800}) - \kappa J(\omega_H^{700})\right\}\right] \tag{6}$$

where

$$\kappa = \left(\frac{\omega_H^{800}}{\omega_H^{700}}\right)^2, \tag{7}$$

$$\beta = \frac{d^2}{2}(1 - \kappa), \tag{8}$$

$$c = \omega_N(\sigma|| - \sigma\perp), \tag{9}$$

$$d = \left(\frac{\mu_0 h}{8\pi^2}\right)\left(\frac{\gamma_H}{\gamma_N}\right)(r_{NH})^{-3}, \tag{10}$$

where $h$ is Planck's constant, $\mu_0$ is the permeability of a vacuum, $r_{NH}$ is the bond length (1.02 Å) and $\sigma|| - \sigma\perp$ is the axial chemical shift tensors for the backbone $^{15}$N nuclei, which is considered $-160$ ppm; and $\gamma_N$ and $\gamma_H$ are the gyromagnetic ratios of $^{15}$N and $^1$H, respectively.

**The secondary structure of the population of the 15-mer**. The secondary structure population of the 15-mer was determined based on the backbone chemical shifts using δ2D[33].

**$^3$J HNHA measurements of the repetitive domains**. 3D $^3$JHNHA experiments[69] were conducted at 10 °C using a 700 MHz Bruker spectrometer equipped with a TCI cryogenic probe and a z-axis gradient coil. The 3D $^3$JHNHA experiments included 16 scans, a 1 s recycle delay, 2048 × 256 × 36 complex points for the $^1$H$^N$, $^1$Hα, N$^H$ dimensions, respectively, and a spectral width of 16 × 11 × 26 ppm, which corresponds to the $^1$H$^N$, $^1$Hα, and N$^H$ dimensions, respectively, using non-uniform sampling (50% NUS). All data were processed using NMRPipe[63] and analyzed using SPARKY[64]. The $^3$JHNHα coupling constant was calculated based on the intensity ratio of the cross peak and the diagonal peak ($I_{cross}/I_{diagonal}$) using the following relationship[69]:

$$I_{cross}/I_{diagonal} = -\tan^2(2\pi^3 J_{HH}\zeta), \tag{11}$$

where $\zeta$ is the delay time of 13.05 ms and $J_{HH}$ is the $^3$JHNHA coupling constant.

**pH effect on repetitive domains**. The backbone chemical shifts of the ($^{13}$C, $^{15}$N) 15-mer were monitored using 2D $^1$H–$^{15}$N HSQC, 2D $^1$H–$^{13}$C HSQC, and 2D projection of H(N)CO spectra from pH 7 to 2.6 at 10 °C using a 700 MHz Bruker spectrometer equipped with a TCI cryogenic probe and a z-axis gradient coil. Each spectrum was recorded using 2048 and 128 complex points for the direct and indirect domain, respectively, and the number of scans was equal to 2, 4, and 8 for 2D $^1$H–$^{15}$N HSQC, 2D $^1$H–$^{13}$C HSQC and 2D projections of H(N)CO, respectively. All data were processed using NMRPipe[63] and analyzed using SPARKY[64].

To obtain information on the $^{15}$N$^H$, $^{13}$C′, $^{13}$Cα and Hα resonances at basic pH, a series of 2D carbon-detected CON (correlates the carbonyl and amide nitrogen) and 2D $^1$H–$^{13}$C HSQC (in the aliphatic region) experiments were performed at 10 °C at three different pH values (7, 9, and 10.2) using a 700 MHz Bruker spectrometer equipped with a TCI cryogenic probe and a z-axis gradient coil. The 2D CON spectra were acquired using 320 scans, a relaxation delay of 1 s, and 128 ($^{15}$N$^H$) and 1024 ($^{13}$C′) complex points with maximum acquisition times of 24.38 and 9.75 ms, respectively.

**Tyr side chain pK$_a$ determination of the repetitive domain**. The Tyr side chain pK$_a$ value was determined by measuring $^{13}$Cε chemical shifts as a function of pH. $^{13}$Cε resonances were assigned based on $^1$H–$^{13}$C HSQC spectra in aromatic regions (carrier position of $^{13}$C at ~125 ppm) from pH 7 to 11.28. $^1$H–$^{13}$C HSQC spectral experiments were conducted at 10 °C using an ECA 600 MHz JEOL spectrometer. All NMR data were processed using NMRPipe[63] and analyzed using SPARKY[64]. All $^{13}$Cε chemical shifts were fit as a function of pH using the Henderson-Hasselbach equation for a single event:

$$\delta_{obs} = \delta_{AH} + \Delta\delta\frac{10^{n_H(pH - pK_a)}}{1 + 10^{n_H(pH - pK_a)}}, \tag{12}$$

where $\delta_{obs}$ is the observed chemical shift as a function of pH, $\delta_{AH}$ is the chemical shift for the protonated form, $\Delta\delta = \delta_{A^-} - \delta_{AH}$ is the change in chemical shift upon deprotonation, and $n_H$ is the Hill coefficient. Calculations were performed using Qtiplot.

**Concentration effect on the repetitive domain**. The 2D $^1$H–$^{15}$N HSQC spectra of the ($^{13}$C, $^{15}$N) 15-mer were recorded as a function of the 15-mer concentration. First, a 2D $^1$H–$^{15}$N HSQC spectrum for ~1 mM ($^{13}$C, $^{15}$N) 15-mer in 10 mM phosphate buffer at pH 7 and 11 °C was recorded using 1024 × 128 complex points of H$^N$ and N$^H$ and four scans. Second, a 2D $^1$H–$^{15}$N HSQC spectrum of ~0.5 mM (twice diluted) ($^{13}$C, $^{15}$N) 15-mer in 10 mM phosphate buffer at pH 7 and 11 °C was recorded using 1024 × 128 complex points for H$^N$ and N$^H$ and 16 scans. All spectra were processed in the same manner using NMRPipe and visualized using SPARKY. The ratio of S/N values of $^1$H–$^{15}$N HSQC spectra of the 15-mer (concentration 1 mM, number of scans = 4) and the 15-mer (concentration 0.5 mM, number of scans = 16) was plotted as a function of the residue number

**Temperature effect on the repetitive domain**. The effect of temperature on the repetitive domain was investigated using NMR and CD spectroscopy. Series of 2D $^1$H–$^{15}$N HSQC and 2D $^1$H–$^{13}$C HSQC spectra of the ($^{13}$C, $^{15}$N) 15-mer at pH 7 were recorded at 10, 15, 20, and 25 °C. Each spectrum was recorded using 2048 and 128 complex points for the direct and indirect domain, respectively, and the number of scans was equal to 4. {$^1$H}–$^{15}$N heteronuclear NOE experiments for the ($^{13}$C, $^{15}$N) 15-mer at 15 °C were recorded and processed in similar manner as the {$^1$H}–$^{15}$N heteronuclear NOE experiment for the 15-mer at 10 °C. All spectra were processed in the same manner using NMRPipe and visualized using SPARKY. CD spectra of ~100 μM 15-mer were recorded at 10, 15, 20, 25, and 30 °C. The CD spectra were recorded from 230 to 190 nm in a 1 mm path length quartz cuvette using a JASCO J-820 spectropolarimeter. The wavelength step for the CD measurements was 1 nm, and the scan rate was 500 nm/min. The spectra shown were subtracted from the background and averaged over 10 consecutive scans.

**Data availability**. The NMR assignments of repetitive domain of spider dragline silk have been deposited in Biological Magnetic Resonance Bank (BMRB) under accession code 27460. All other data are available from the authors upon reasonable request.

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

## Acknowledgements

This work was financially supported by the Impulsing Paradigm Change through Disrupt Technologies Program (ImPACT) to K.N. We thank Dr. Atsuya Muranaka from the Advanced Element Research Team, RIKEN Center for Sustainable Resource Sciences for his technical support with setting up the VCD experiments.

## Author contributions

N.A.O. and K.N. designed the research. K.N. and D.L.K. designed the repetitive domain genes. K.N. constructed all plasmid DNA containing repetitive domain genes of spider silk proteins. N.A.O. expressed, purified repetitive domain, and labeled ($^{13}$C,$^{15}$N) the repetitive domains. N.A.O. performed the CD experiments and analyzed the CD data. N.A.O. and A.M. performed the NMR experiments. N.A.O. and F.H. analyzed the NMR data. A.D.M. performed the sequence analysis and analyzed the CD data. N.A.O. and K.N. performed the VCD spectroscopy. N.A.O. and K.N. analyzed the whole data in the context of the mechanism of repetitive domain assembly of spider silk protein and β-sheet formation. N.A.O made all figures. All the authors prepared the manuscript.

## Additional information

**Competing interests:** The authors declare no competing interests.

