## [Peer Review File · Nature Communications]

Reviewers' Comments:

Reviewer #1 (Remarks to the Author)

Oktaviani et. al present a conformational and dynamic study on the repetitive region of recombinant spider silk proteins of varying length using CD and protein solution NMR methods. The results of the paper are quite interesting and publishable following major revisions detailed below. However, even if the major revisions are addressed, I do not see why the results warrant publication as a rapid Nature Communication? Even if the comments below are addressed, the paper mainly confirms the results of previous studies with Raman, NMR and CD on native silk proteins and indicate that they are primarily random coil. All perturbations observed by the authors by NMR are very small and within some of the errors reported. Even if the authors can address some of the comments regarding the determination of the PPII helix, proposing the presence of a small population of PPII helix is in agreement with previous Raman studies. So again, not sure why a rapid Nature Communication.

Recommendation: Major Revisions and submit to lower tier journal.

Comments:

(1) What about temperature? The authors make many comparisons to previous studies with NMR, CD and Raman on native silk proteins. However, most, if not all of these experiments were conducted at room temperature. All the measurements in this paper were conducted at 10 C. Have the authors made room temperature measurements and if so, do they see the small NMR perturbations at room temperature as well? The dynamics are anticipated to slow down a bit as the temperature is lowered and could be the cause of the small NMR perturbations observed.

(2) The authors report conventional solution NMR relaxation measurements T2 and NOE to determine dynamics. However, no T1 measurements were reported. Why not? It is typical to report the T1, T2 and NOE.

(3) What is the error in the NOE measurement presented in Figure 3? One can view that result as an average NOE value of -0.2 ± 0.1 . And thus, they are essentially the same. Either way a slightly positive NOE at 700 MHz still indicates rapid ns backbone dynamics and a high degree of flexibility for all the residues which is again in line with the random coil structure.

(4) The strongest NMR evidence for the PPII structure are the 3-bond J-coupling measurements. However, it is not clear if these measurements were made for the other amino acids in the glycine-rich region. How do they compare to the PPII structure. The PPII structure exhibits similar chemical shifts to random coil and similar 3-bond J-couplings for the most part. Within the error of the measurement the random coil and PPII or similar. Comparisons between the PPII helix and random coil three bond J-couplings for the other non-Gly amino acids would go a long way in removing this ambiguity (e.g. Gln, Tyr, Leu with value reported separately).

(5) This reviewer is not convinced that these small effects in the NMR are not due to entanglement for the higher MW recombinant silk proteins. At the concentrations studied, entanglement will have more of an impact on the higher MW protein. The authors discuss no change in chemical shifts at the different concentrations but, what about the dynamics? And the 3-bond J-couplings? No change in the chemical shifts is consistent with a random coil but, what about the other measurements?

(6) The authors claim that because the repetitive domain displays no pH effect (conversion to β -sheet at acidic pH) then, the C and N-termini must be needed for this pH dependent β -sheet

formation. Even if this is the case, one would need to engineer the repetitive domain with the C/N-termini and show that it did convert to β -sheet when the pH was changed. This experiment has not been done. And thus, this discussion is completely speculative. There could be countless reasons for why the repetitive domain does not convert to β -sheet at acidic pH that have nothing to do with the presence of the termini or not.

Reviewer #2 (Remarks to the Author)

Manuscript "Conformation and dynamics studies of the soluble precursor repetitive domain of spider elucidate initiation step of β -sheet formation" by Oktaviani et al, represent interesting structural inside on spider silk core domain. In this work author used several artificial model peptides comprising different number of repetition units related to natural MaSp protein of *N. clavipes*. The peptides we studied using plethora of advanced solution NMR technique to elucidate the role of structural unit repetition on chain dynamic and population of secondary structures. The peptides were also studied at different pH, and concentration conditions, both factors playing important roles in the natural spinning process. Manuscripts presents novel and interesting findings on secondary structure distributions and flexibilities along the repetitive unit as well as on intra-chain interactions between the units. This findings allow proposal of a new structural model of the soluble spider silk domain, which exhibits pre-structured (prefibrillar) silk population, which might facilitated fast beta-sheet formation during the fiber spinning. They show also that the pH changes in the range, which occurs in the spinning duct, do not influence this pre-structuring. These findings represent important contribution to understanding how the highly soluble proteins in the spider dope transform into insoluble structures in the though fiber. Nevertheless, there are several major and minor issues, author should correct and comment, before publication in Nature Communication

Introduction:

Minor comments:

Unclear statement: "Several studies indicated that the NTD and CTD display strong pH dependence and are essential...." Ph dependence of what? Structures? Interactions? Please clear this.

The placement of the statement: "A recent report also showed the importance of tyrosine-tyrosine interaction in repetitive domain of silkworm silk to promote β -sheet formation" in the section about CTD and NTD is confusion; it might be more suitable in the section with pH changes.

Results

Major comments:

CD Spectroscopy (Figure 1.)

A) Poor quality of the spectra from several points of view:

- Wong scale or unit of molar ellipticity on Y-axis. Typical minimum at 200 nm should be at -5000 to -10000 deg.cm²/dmol. The intensities here are actually 100 times lower, than expected.
- Unexpectedly high CD intensity of the dimer at 200 nm in comparison to monomer and trimer points to rather miscalculation of the peptide concentration. Please correct this.
- The spectra of the 15-mer are claimed to be similar to native protein in ref. 25, but neither the intensity nor the profile are similar. Authors omit any comments to the unusual peak at 230 nm, which is very untypical for such kind of proteins. I suspect some impurities in this sample. Please check this.

B) Authors should consider the re-evaluation of the CD spectra. Additionally, I found this CD section only supportive for the whole story, and it would be more suitable as supporting information.

Section on pH effects (Figure 4).

I found experiments and comments to the titrations in Figure 4 are not relevant for the story, since these experiments follow the backbone and residue changes at pH 7 – 11 i.e. conditions not related to natural spinning dope or spinning process, which is indeed in focus of the paper. Authors should focus rather on the titration and changes below pH 7 as described in Figure 5.

Discussion

Major comments:

Figure 6; Please revise the sentence in the legend: "The polyalanine region is blue, and the glycine-rich region is red." To the sentence: "The polyalanine region is red, and the glycine-rich region is blue". It is obvious and generally well described that the polyalanines undergo the structural transformation into beta sheets, whereas PPII helix either not change or turn to alpha-helix, as described also in this manuscript.

More details could be included into Figure 6B: schematic formation of the PPII helices on the right site and schematic intra-chain interaction in the PPII regions.

Minor comments:

Recent NMR studies on natural and recombinant spider silk protein and the influence of pH on the polypeptide chain structure also should be considered in the discussion:

Leclerc, J., Lefèvre, T., Pottier, F., Morency, L.-P., Lapointe-Verreault, C., Gagné, S. M. and Auger, M. (2012), Structure and pH-induced alterations of recombinant and natural spider silk proteins in solution. *Biopolymers*, 97: 337–346.

Leclerc, J., Lefèvre, T., Gauthier, M., Gagné, S. M. and Auger, M. (2013), Hydrodynamical properties of recombinant spider silk proteins: Effects of pH, salts and shear, and implications for the spinning process. *Biopolymers*, 99: 582–593.

Supporting Information:

Major comment:

Supplementary Figure 2:

B) And C) lettering is missing in the figure

Wrong description of Y-axes: Author claim in the legend to present ^1H - ^{15}N -HSQC spectra, however, chemical shifts on the Y-axes are described as ^{13}C . Please correct this.

Legend C): claimed aliased peak form 43.89 ppm, which however cannot be found in the spectra. Please correct this.

The scale of the X-axes in B and C are the same (8.8-7.8) but the dimension of the axes differ. Please adjust.

Manuscript ID: NCOMMS-16-28855

Title: Conformation and dynamics studies of the soluble precursor repetitive domain of spider silk to elucidate the initial mechanism of β -sheet formation

Corresponding author: Keiji Numata, RIKEN Center for Sustainable Resource Science, Japan

RESPONSES TO REVIEWER COMMENTS

Reviewer #1:

Comment 1

Oktaviani *et. al* present a conformational and dynamic study on the repetitive region of recombinant spider silk proteins of varying length using CD and protein solution NMR methods. The results of the paper are quite interesting and publishable following major revisions detailed below. However, even if the major revisions are addressed, I do not see why the results warrant publication as a rapid Nature Communication? Even if the comments below are addressed, the paper mainly confirms the results of previous studies with Raman, NMR and CD on native silk proteins and indicate that they are primarily random coil.

Response

This study reveals the detailed conformation and dynamics of the repetitive domain of spider silk proteins and clearly explains the initiation step of β -sheet formation of spider silk proteins, as well as the rationale for the high solubility of spidroin in the major ampullate gland. Therefore, we consider this study to be important for understanding the initial β -sheet formation of spider silk proteins, which to date remains poorly understood. A previous study reported the presence of the PII helix population as well as the random coil population in soluble spider silk proteins (Ref 23. Lefèvre, et al, Biomacromolecules, 2007). However, the following issues must be addressed in terms of the conformation and dynamics of the soluble precursor repetitive domain of spider silk proteins in order to fully understand the mechanisms involved.

Issue 1 - characterization methods

Raman and far-UV circular dichroism (CD) spectra provide structural information regarding the overall sequences, whereas NMR can provide the local structures of each amino acid in the overall sequences. To date, several NMR studies have reported the presence of soluble spider silk proteins, which are present only in the random coil population (Ref 24. Hijirida et al, Biophys J, 1996; Ref 13. Jenkins et al, Soft Matter, 2012; Ref 25. Xu et al, Polymer, 2014). These results of Refs 13, 24 and 25 differ from other spectroscopy results (Ref 23. Lefèvre et al, Biomacromolecules, 2007).

In this study, we used **3 different types of spectroscopy**, multi-dimensional NMR, CD and vibrational circular dichroism (VCD), to determine the conformation and dynamics of the soluble repetitive domain. A combination of CD and VCD spectroscopies is useful for revealing the PII helix population in the proteins (Bochicchio and Tamburro, Chirality 14:782–792, 2002). Furthermore, multidimensional NMR data provide detailed conformation and dynamics information

for the repetitive domain.

Here, we showed that the conformation and dynamics of the soluble precursor repetitive domain of spider silk provides a fundamental explanation for the initiation step of β -sheet formation. Our data clearly demonstrated that the soluble repetitive domain of spider silk protein consisted of 2 populations: random coil (~65%) and PPII helix (~24%). The PPII helix population is distributed over the glycine-rich region, whereas the random coil population is distributed over the polyalanine region. This finding is consistent with dynamics data, which have consistently shown that the glycine-rich region has more limited flexibility compared with the polyalanine region.

Issue 2 - silk protein domains

Many recent studies have primarily focused on the structure and role of highly conserved N- and C-terminal domains (NTD and CTD) of spider silk proteins (Ref 14. Askarieh et al, Nature, 2010; Ref 15. Hagn et al, Nature 2010; Ref 16. Hagn et al, Angew. Chem. Int. Ed., 2011; Ref 17. Jaudzems et al, J. Mol Biol, 2012, Ref 18. Gauthier et al, Biomacromolecules, 2014). However, the NTD and CTD are relatively smaller than the repetitive domain of spider silk protein. Thus, those studies do not explain why the β -sheet can be formed in the absence of NTD and CTD (Ref 20. Xia et al, PNAS, 2010; Ref 21. Heidebrecht et al, Adv Material, 2015; Ref 22. Humenik et al, J Struc Biol, 2014).

To clarify this issue, we used a combination of solution NMR, CD and VCD to analyze the repetitive domain of the spider silk proteins because it dominates the length of protein chain. We used the recombinant repetitive domain of spidroin of different lengths to analyze the length effect on the conformation and dynamics of the repetitive domain. We investigated **the effect of the length of the repetitive domain as well as the pH, temperature and concentration effects on the conformation and dynamics of the repetitive domain**. Based on our results, we found that the conformation and dynamics of multi repeats of the repetitive domain were different from that of the monomer because of the presence of the PPII helix population, which then undergoes intramolecular interactions by forming a reverse turn.

Important findings in this study

Compared with previous studies, by using multiple characterization methods and the repetitive domain of spider silk protein, this study provides **two novel and important findings** that clearly explain the initiation step of β -sheet formation of spider silk proteins as follows.

Finding 1

Our study showed that the repetitive domain of spider silk proteins consists of two major populations: random coil (65%) and PPII helix (24%). As depicted in **Fig. 3**, approximately 25% of the total secondary structure population in the glycine-rich region of the 15-mer is the PPII helix population, whereas approximately 80% of the total secondary structure population in the polyalanine region is the random coil population. This finding was supported by several lines of evidence:

1. Our NMR data demonstrated that the repetitive domain (15-mer) is not fully random coil protein. The most direct evidence was shown by the more positive NOE value of the 15-mer compared with the monomer. This result strongly indicated that the 15-mer had more limited flexibility compared with the monomer. Furthermore, a more comprehensive analysis using ^{15}N relaxation and $\{^1\text{H}\}$ - ^{15}N heteronuclear NOE of the 15-mer in two different magnetic fields (700 MHz and 800 MHz) revealed that the spectral density at 0 frequency $J(0)$ of 15-mer was approximately 5.5 ns, which was longer than the $J(0)$ of the monomer at approximately 2.2 ns.
2. For a fully random coil protein, the additional repetitive domain would only affect the backbone chemical shift of residues $i-1$ and $i+1$. However, the observation that more distant residues ($^{26}\text{GLGSQG}^{31}$) were influenced by the addition of a more repetitive domain (Supplementary Fig. 5 and 6) suggested that the repetitive domain was not a fully random coil protein. Furthermore, as the number of repetitive domains increased, the helical propensity at the GLGSQGTS region increased, suggesting that the PPII helix population was stabilized by a higher number of repetitive domains (Supplementary Fig 3). The PPII helix stabilization on the repetitive domain may have resulted from limited flexibility, particularly in the GLGSQGTS region. Furthermore, this PPII helix stabilization in the presence of a higher number of repetitive domains enables spidroin to readily form intramolecular interactions and explains why stronger spider silk fibers can be achieved when the silk protein contains many repetitive domains (ref 20 Xia et al, PNAS, 2010).
3. The distribution of the secondary structure of the repetitive domain demonstrated that approximately 25% of the PPII helix conformation was distributed over the glycine-rich region, whereas the polyalanine region contained a greater random coil population. This result was corroborated by the observation that the glycine-rich region has more limited flexibility than the polyalanine region, as demonstrated by $\{^1\text{H}\}$ - ^{15}N heteronuclear NOE.
4. The CD spectra of the monomer only showed minima at 200 nm and revealed a typical pattern of the random coil structure, whereas the dimer, trimer, hexamer and 15-mer showed minima at 196 nm and a weakly positive band at 215 nm, indicating the presence of the PPII helix population (**Fig. 1B**). These findings are supported by the observation that the glycine-rich region of the repetitive domain had more limited flexibility than the polyalanine region based on our NMR results.
5. The VCD spectra of the 15-mer in D_2O at pH 7 and 25°C showed that the 15-mer had distinctive maxima at 1662 cm^{-1} and observable minima at 1636 cm^{-1} , thus representing a typical of VCD spectrum containing the PPII helix population. In contrast, the delta absorbance of the VCD spectrum of the

monomer significantly decreased, which demonstrated a larger population of random coil proteins.

These details and explanations have been added to the manuscript as follows.

Page 2, line 38-42

Abstract

“The soluble repetitive domain contains two major populations: ~65% random coil and ~24% polyproline type II helix (PPII helix). The PPII helix conformation in the glycine-rich region is proposed as a soluble prefibrillar region that subsequently undergoes intramolecular interactions by forming a reverse turn.”

Page 5, line 108-114

Introduction

“Our results demonstrate that the repetitive domain of dragline silk consists of two major populations: random coil (~65%) and PPII helix (~24%). We propose that the PPII helix population in the glycine-rich region might serve as the prefibrillar form of spider dragline silk and contribute to the efficiency of the spinning process because the PPII helix can easily undergo intramolecular interactions in response to shear forces and dehydration by forming a reverse turn²⁸. Additionally, the limited flexibility of the glycine-rich region might function to maintain the solubility of spider silk in the ampullate gland.”

Page 6, line 134-152

Structural propensity of the repetitive domains at low and high concentrations by CD and VCD spectroscopies

“The structural propensities of the repetitive domain of spider silk at low concentration (10–20 μM) were determined via CD spectroscopy. The monomer spectra had a minimum at approximately 200 nm, which is typical of an intrinsically disordered protein^{29,30}. In contrast, the spectra of the dimer, trimer, hexamer and 15-mer demonstrated minima at approximately 196 nm; in particular, the CD spectra of the hexamer and 15-mer displayed small but detectable maxima at approximately 215 nm (**Fig. 1B**), which is consistent with the PPII helix conformation^{29,30}. Importantly, similar minima were detected in the CD spectra of multiple repetitive domains (dimer, trimer, hexamer and 15-mer) as the spectra of native spider silk proteins from *Nephila edulis* in fresh conditions (incubation time $t=0$)³¹.

The presence of the PPII helix population of the repetitive domains was evaluated in D_2O at a high concentration (8 mM) via VCD spectroscopy in the amide I region corresponding to the carbonyl stretch. The VCD spectrum of the 15-mer at pH 7 and 25°C in the amide I region demonstrated a maximum at 1662 cm^{-1} and an observable minimum at 1636 cm^{-1} (**Fig. 1C**), suggesting the presence of the PPII helix^{23,32}. In contrast, the delta absorbance of the VCD spectrum of the monomer decreased significantly, indicating that the monomer predominantly contains the unordered random coil population rather than the PPII helix population³³.”

Page 10, line 221-259

Backbone dynamics of the repetitive domains

“The backbone dynamics of the monomer and 15-mer were probed with the $^{15}\text{N}\text{T}_1$, $^{15}\text{N}\text{T}_2$ relaxation and $\{^1\text{H}\}$ - ^{15}N heteronuclear nuclear Overhauser effect (NOE)

under two different magnetic fields (700 MHz and 800 MHz) (**Supplementary Fig. 7**). The average $^{15}\text{NT}_1$ relaxation times of the 15-mer were 538 and 794 ms at 700 and 800 MHz, respectively, whereas the average $^{15}\text{NT}_1$ relaxation times of the monomer were 416 and 420 ms at 700 and 800 MHz, respectively (Supplementary Fig. 7A and 7D). Furthermore, the average $^{15}\text{NT}_2$ relaxation times of the 15-mer at 700 and 800 MHz were 183 and 134 ms, respectively, which was shorter than the $^{15}\text{NT}_2$ relaxation times of the monomer, which were 189 and 169 ms at 700 and 800 MHz, respectively. The shorter $^{15}\text{NT}_1$ and longer of $^{15}\text{NT}_2$ relaxation times of the monomer suggested that the monomer had a shorter rotational correlation time and faster dynamics in comparison to the 15-mer³⁷.

The local dynamics (individual H-N bond vector) of the monomer and 15-mer were probed with $\{^1\text{H}\}$ - ^{15}N heteronuclear NOE experiments using 2 magnetic fields (700 and 800 MHz) (Supplementary Fig. 7C and 7F). Our results demonstrated that the 15-mer had higher NOE values at both 700 and 800 MHz NMR than the monomer, suggesting that the 15-mer was less flexible than the monomer. Furthermore, our dynamics data revealed higher NOE values of the 15-mer in the glycine-rich region than in the polyalanine region (**Fig. 3A**).

The $^{15}\text{NT}_1$, $^{15}\text{NT}_2$ $\{^1\text{H}\}$ - ^{15}N heteronuclear NOE data for the monomer and 15-mer using 2 different magnetic fields, 700 and 800 MHz, were then translated into spectral densities, which revealed the molecular motion at certain angular frequency. Those spectral densities of the 15-mer and monomer were calculated at $J(0)$, $J(\omega\text{N})$ and $J(0.87\ \omega\text{H})$, which corresponded to 0, 70, 80, 609 and 696 MHz (**Supplementary Fig. 8**). The spectral densities $J(609\ \text{MHz})$ and $J(696\ \text{MHz})$ of the monomer were longer (~ 0.03 - $0.04\ \text{ns/rad}$) than those of the 15-mer (~ 0.01 - $0.02\ \text{ns/rad}$) (Supplementary Fig. 8A). Furthermore, the spectral densities $J(70\ \text{MHz})$ and $J(80\ \text{MHz})$ of the monomer were not significantly different, whereas the spectral density $J(80\ \text{MHz})$ of the 15-mer decreased in comparison to its spectral density at $J(70\ \text{MHz})$ (Supplementary Fig. 8B). These results also indicated that the monomer underwent faster motion than the 15-mer^{38,39}.

The most striking results were obtained for the spectral densities of the 15-mer and monomer at a frequency of 0 MHz ($J(0\ \text{MHz})$) (Supplementary Fig. 8C) because this spectral density reflects the rotational correlation times (τ_c) of these molecules. The $J(0)$ values of 15-mer ($\sim 5.5\ \text{ns}$) suggested that the rotational correlation time of the 15-mer was longer than the rotational correlation time of the monomer, which presented a $J(0)$ of approximately 2.2 ns. This finding also indicated that the 15-mer underwent slower motion compared with the monomer. Taken together, all the dynamics data indicated that the 15-mer did not behave as a fully random coil protein because it rotated more slowly than intrinsically disordered proteins, which normally have a very short spectral density $J(0)$ (0.5-2.2 ns)³⁹⁻⁴¹.

Page 11, line 264-267

“The larger population of the PPII helix (approximately 25% of the total secondary structure population) was attributed to the glycine-rich region, with a higher probability observed near Gln. In the polyalanine region, the PPII helix population decreased and the random coil population increased.”

Page 18, line 405-419

“Furthermore, a previous study using solid-state NMR indicated that the conserved LG(G/S)QG motif in the glycine-rich region (**Supplementary Fig. 15**) adopts a turn conformation in MaSp1 fibers⁵⁹. Such findings are supported by our results, which showed that the structural propensity (**Supplementary Fig. 3**) and backbone carbonyl chemical shifts in the GLGSQGTS region differ from those of the monomer and other repetitive domains (**Supplementary Fig. 5 and 6**). These results suggest that the chemical environment of this motif in multiple repetitive domains (dimer, trimer, hexamer and 15-mer), which are able to undergo intramolecular interactions, differs from the chemical environment of this motif in the monomer. Our data also demonstrate that when the number of repetitive domains is increased, the helical propensity of the GLGSQGTS region also increases, suggesting that the PPII helix population is stabilized by the greater number of repetitive domains (**Supplementary Fig. 3**). PPII helix stabilization in the repetitive domain may be related to the limited flexibility, particularly in the GLGSQGTS region. Furthermore, PPII helix stabilization with a larger number of repetitive domains enables spidroin to readily form intramolecular interactions and explains why stronger spider silk fibers can be achieved when the silk protein contains high numbers of repetitive domains²⁰.”

Page 22, line 479-483

“In conclusion, the soluble repetitive domains of spider dragline silk at neutral pH have two major populations: PPII helix (24%) and random coils (65%). The PPII helix conformation is more prevalent in the glycine-rich regions, which present less flexibility than the polyalanine region, which contains more random coils. The glycine-rich region with the PPII helix population is proposed as the soluble prefibrillar form, which readily supports the transformation into insoluble silk fiber.”

Finding 2

Our results showed that the structure and assembly of the repetitive domain is pH independent, which indicates that pH changes that occur in the spinning duct do not affect the conformation of the pre-fibrillar repetitive domain. In addition, we also clarified that the pK_a of the tyrosine side chain in the repetitive domain is approximately 10.3; hence, the solubility of the repetitive domain is not determined by deprotonation of Tyr as previously suggested (Ref 55: Exler *et al.* Angew. Chem. Int. Ed., 2007). Moreover, in this study, we consider the glycine-rich region to have limited flexibility because of the larger PPII helix population, which is an essential factor for maintaining the solubility of spider silk by preventing intramolecular interactions of hydrophobic sequences. These explanations for Finding 2 have been added to the main text as follows.

Page 14, line 294-299

“To understand the pH effect on the structure of the repetitive domain at an acidic pH, several 2D NMR experiments, namely ¹H-¹⁵N HSQC, ¹H-¹³C HSQC and 2D projection of H(N)CO, were performed to follow the backbone (C α , H α , N^H, and C^{\prime}) chemical shifts of the 15-mer from pH 2.6 to 7.0. We found that neither chemical shift changes nor line broadening occurred in that pH range (**Fig. 4**). This result implies that the repetitive domains of MaSp1 do not undergo any conformational changes in response to a decrease in pH.”

Page 15, line 308-322

“The pH effect on the repetitive domain at a basic pH was probed using 2D CON and 2D ^1H - ^{13}C HSQC aliphatic experiments. The 2D CON spectrum is useful for examining the conformation details of an unfolded protein because this spectrum displays good dispersion of the carbonyl and amide nitrogen chemical shifts. In addition, both carbonyl and amide nitrogen are non-labile nuclei; therefore, they do not exchange water proton signals at a high pH⁴⁹. Our pH titration data from pH 7 to pH 9 did not show any chemical shift change in the 2D CON or 2D ^1H - ^{13}C HSQC aliphatic spectra of the 15-mer (**Supplementary Fig. 10A and B**). After further pH titration under basic conditions, only marginal chemical shift changes were observed around Tyr residues at pH 10.2. This result can be explained by deprotonation of the Tyr side chain. Accordingly, deprotonation of the Tyr side chain only led to a small enhancement of the helical propensities around the Tyr residue (**Supplementary Fig. 11**). We further determined the tyrosine side chain pK_a using $^{13}\text{C}\epsilon$ as a chemical shift reporter (**Supplementary Fig. 10C**). Our pH titration data (pH 7.0–11.3) indicated that the pK_a of the Tyr side chain of the 15-mer was 10.25 ± 0.02 (**Supplementary Fig. 10D**), which is identical to the typical value determined for solvent-exposed Tyr⁵⁰.”

Page 21-22, line 450-476

“Our investigation of the acidic pH effect revealed no conformational changes in the repetitive domains, suggesting that pH changes in the spinning gland do not influence the prefibrillar region of the repetitive domain. The conformational change in a protein as a function of pH is normally accompanied by electrostatic interactions among acidic and basic groups, which involves protonation and deprotonation events. Because acidic groups were not observed in the repetitive domain, no conformational changes were observed at acidic pH. However, in vitro studies of native and recombinant spider dragline silk composed of NTD, repetitive domains and CTD demonstrated β -sheet formation at an acidic pH⁶⁰⁻⁶². Thus, based on the results of the present and previous studies, we concluded that CTD and NTD might be necessary for inducing β -sheet formation in spider silk during the early stage of the spinning process. Furthermore, dehydration, shear forces, and changes in biochemical environment are also involved in β -sheet formation of spider silk proteins.

Previously, volumetric and spectrophotometric titration studies of native spidroin (*Nephila edulis*) demonstrated two apparent pK values: 6.74 and 9.21⁶³. The first pK value was hypothesized to originate from the deprotonation of different residues, whereas the latter pK value was thought to be related to the deprotonation of Tyr side chain residues⁶³. In addition, deprotonation of the Tyr side chain has been thought to be involved in the solubility of spider dragline silk at pH values greater than 8.5⁶⁴. Upon pH titration in the basic pH range in this study, chemical shift changes were not observed for the repetitive domain except for a slight change at pH 10.2 because of deprotonation of the Tyr side chain ($pK_a=10.3$). Although the Tyr side chain pK_a of the repetitive domain was determined in the absence of salts (NaCl, KCl), the presence of high concentrations of salt (approximately 0.4 M) only affected amino acid pK_a values ranging from approximately 0.2–0.9 pH units as previously reported⁶⁵. Based on the pK_a value of the Tyr side chain of the repetitive domain, deprotonation of the Tyr side chain was not essential for increasing the solubility of spider dragline silk because the pK_a value of the Tyr side chain at pH

10.3 is too basic for the ampullate gland.”

According to the two findings mentioned above, **we consider this study to provide important insights for a better understanding of β -sheet formation of spider silk proteins via PPII helix conformation in the glycine-rich region.** We hope that the reviewer agrees that our findings provide novel and fundamental knowledge for scientists in many different research fields, particularly in the field of structural proteins.

Comment 2

All perturbations observed by the authors by NMR are very small and within some of the errors reported. Even if the authors can address some of the comments regarding the determination of the PPII helix, proposing the presence of a small population of PPII helix is in agreement with previous Raman studies. So again, not sure why a rapid Nature Communication

Response

As indicated by the reviewer, the backbone chemical shifts of repetitive domains are not extremely elevated relative to typical random coil values. This is because these data are based on the secondary structure distribution of the repetitive domain (15-mer), and the PPII helix population in the glycine-rich region is only 25% of the total secondary structure population. However, based on Supplementary Fig. 3, we found that when the number of repetitive domains increased, the helical propensity also increased, particularly in the GLGSQGTS region. The GLGSQG region is located in the glycine-rich region and is highly conserved (Supplementary Fig 15), and it adopts a turn conformation in MaSp1 fibers. This finding suggests that the PPII helix population is stabilized by the larger number of repetitive domains. This PPII helix stabilization in the repetitive domain may be related to the limited flexibility, particularly in the GLGSQGTS region. Furthermore, this PPII helix stabilization in the presence of a greater number of repetitive domains enables spidroin to readily form intramolecular interactions by forming a reverse turn, and it also explains why stronger spider silk fibers were achieved when the silk proteins contained a large number of repetitive domains (ref 20. Xia et al, PNAS, 2010).

Furthermore, the merits of this study are not only related to the determination of the PPII helix as well as random coil populations for the repetitive domain but also the detailed information provided for the PPII helix. **Our study clearly demonstrated that the distribution of the PPII helix population was higher in the glycine-rich region,** which was also supported by the more limited flexibility of the glycine-rich region compared with the polyalanine region. Thus, the detailed molecular conformation and dynamics provide a fundamental explanation for the initiation step of β -sheet formation. We have added an additional explanation in the Discussion as follows:

Page 19, line 405-419

“Furthermore, a previous study using solid-state NMR indicated that the conserved LG(G/S)QG motif in the glycine-rich region (**Supplementary Fig. 15**) adopts a turn conformation in MaSp1 fibers⁵⁹. Such findings are supported by our results,

which showed that the structural propensity (**Supplementary Fig. 3**) and backbone carbonyl chemical shifts in the GLGSQGTS region differ from those of the monomer and other repetitive domains (**Supplementary Fig. 5 and 6**). These results suggest that the chemical environment of this motif in multiple repetitive domains (dimer, trimer, hexamer and 15-mer), which are able to undergo intramolecular interactions, differs from the chemical environment of this motif in the monomer. Our data also demonstrate that when the number of repetitive domains is increased, the helical propensity of the GLGSQGTS region also increases, suggesting that the PPII helix population is stabilized by the greater number of repetitive domains (**Supplementary Fig. 3**). PPII helix stabilization in the repetitive domain may be related to the limited flexibility, particularly in the GLGSQGTS region. Furthermore, PPII helix stabilization with a larger number of repetitive domains enables spidroin to readily form intramolecular interactions and explains why stronger spider silk fibers can be achieved when the silk protein contains high numbers of repetitive domains²⁰.”

Page 19, line 421-426

“Although the PPII helix population of the 15-mer represents only 25% of the total secondary structure population in the glycine-rich region, the presence of PPII helix populations in the soluble repetitive domains plays an important role in the spinning efficiency because this conformation readily undergoes intramolecular interactions. Here, we propose that soluble spider silk occurs in equilibrium with two major populations (**Schematic illustration in Fig. 5**): random coils (65%) and PPII helix (24%).”

Comment 3

What about temperature? The authors make many comparisons to previous studies with NMR, CD and Raman on native silk proteins. However, most, if not all of these experiments were conducted at room temperature. All the measurements in this paper were conducted at 10 C. Have the authors made room temperature measurements and if so, do they see the small NMR perturbations at room temperature as well? The dynamics are anticipated to slow down a bit as the temperature is lowered and could be the cause of the small NMR perturbations observed

Response

According to the comment, we have performed additional NMR and CD spectroscopy experiments to clarify the temperature effect on the conformation and dynamics of the repetitive domain of spider silk proteins (**Supplementary Fig. 13**). Based on the NMR data, the ¹H-¹⁵N HSQC spectra showed decreased signal intensities at higher temperatures (15°C and 20°C) because of the rapid exchange of amide protons with water. However, the overlay of the C α -H α region from ¹H-¹³C HSQC spectra clearly demonstrated no chemical shift differences at higher temperatures. These results suggested that there were no significant changes in the conformation of the soluble repetitive domain.

Furthermore, the dynamics data showed that the overall repetitive domain of spider silk proteins was more flexible at a higher temperature (15°C), although the glycine-rich region consistently showed more limited flexibility compared with the polyalanine

region. Consistent with the NMR data, the CD data for the 15-mer at different temperatures (15°C, 20°C, 25°C and 30°C; note, this covers ranges of temperature where spiders would be active) also consistently demonstrated minima at 196 nm and a weaker positive band at 215 nm, indicating the presence of the PPII helix population at those elevated temperatures (**Supplementary Fig. 13D**). However, the decrease in ellipticity of the 15-mer at approximately 215 nm at higher temperatures (**Supplementary Fig. 13D**) indicated that the PPII helix population for the repetitive domain decreased as the temperature increased, which is also characteristic of PPII helix model peptides (ref 51).

To explain these results, we have added an additional paragraph in the Results as follows.

Page 16-17, line 340-355

“Temperature effect on the repetitive domain

The effect of temperature on the conformation and dynamics of the soluble repetitive domain were investigated using NMR and CD. As the temperature increased from 10°C to 25°C, the signal intensities of the ¹H-¹⁵N HSQC spectra decreased because of the rapid exchange of amide protons to water (**Supplementary Fig. 13A**). However, based on the overlay of the C α -H α region from the ¹H-¹³C HSQC spectra, no chemical shift changes were observed from 10°C to 25°C (**Supplementary Fig. 13B**), suggesting that there were no significant changes in the conformation of the soluble repetitive domain. The {¹H}-¹⁵N heteronuclear NOE data for the repetitive domain consistently showed that the glycine-rich region had more limited flexibility compared with the polyalanine-region, even at a higher temperature (15°C) (**Supplementary Fig. 13C**). Furthermore, the CD spectra of the 15-mer at different temperatures (**Supplementary Fig. 13D**) also demonstrated minima at 196 nm and weak maxima at approximately 215 nm, which persisted from 10°C to 30°C, suggesting the presence of the PPII helix population²⁹. However, the decrease in the ellipticity of the 15-mer at approximately 215 nm at higher temperatures (**Supplementary Fig. 13D**) indicated that the PPII helix population of the repetitive domain decreased as the temperature increased⁵¹.”

We also have added **Supplementary Fig. 13** as follows.

Supplementary Figure 13: Temperature effect on the repetitive domain. (A) ^1H - ^{15}N HSQC spectra of the 15-mer from 10°C until 25°C. As the temperature increased, the signal intensities of the amide proton of the 15-mer decreased due to the rapid exchange of the amide proton signal with water. (B) Overlay of $\text{C}\alpha$ - $\text{H}\alpha$ region of ^1H - ^{13}C -HSQC of the 15-mer at different temperatures. Blue is 10°C, green is 15°C, pink is 20°C and red is 25°C. No changes of $\text{C}\alpha$ - $\text{H}\alpha$ chemical shifts of 15-mer at different temperatures were observed. This finding indicates that conformational changes of the 15-mer do not occur at different temperatures. (C) $\{^1\text{H}\}$ - ^{15}N heteronuclear NOE of the 15-mer at 10°C and 15°C. The glycine-rich region consistently displayed limited flexibility compared with the polyalanine region (in red shadow) at different temperatures. (D) CD spectra of the 15-mer from 10°C up to 30°C. CD spectra of the 15-mer at different temperatures always showed minima at 196 nm and weak positive maxima at approximately 215 nm, which indicates that the 15-mer always contains PPII helix populations at different temperatures. However, as shown in the insert of Fig. 13D, the ellipticity of the 15-mer near wavelength 215 nm decreased at higher temperatures, which suggested that the PPII helix population decreased when the temperature increased.

Comment 4

The authors report conventional solution NMR relaxation measurements T2 and NOE to determine dynamics. However, no T1 measurements were reported. Why not?

It is typical to report the T1, T2 and NOE

Response

$^{15}\text{N}T_2$ relaxation and $\{^1\text{H}\}$ - ^{15}N heteronuclear NOE are the most sensitive experiments for probing changes in the motional dynamics of unfolded proteins, as reported previously (Konrat Magn Reson, 2014). However, as pointed out by the reviewer, we have added $^{15}\text{N}T_1$ relaxation data for the 15-mer at 10°C to complete all the relaxation data. In addition, we also measured the $\{^1\text{H}\}$ - ^{15}N heteronuclear NOE and $^{15}\text{N}T_1$ and $^{15}\text{N}T_2$ relaxation of the 15-mer and monomer using two different magnetic fields, 700 MHz and 800 MHz. Based on the data, we calculated the spectral densities using different magnetic fields. For these dynamics data, we have added an additional explanation in the Results section as follows:

Page 10, line 221-238

Backbone dynamics of the repetitive domains

“The backbone dynamics of the monomer and 15-mer were probed with the $^{15}\text{N}T_1$, $^{15}\text{N}T_2$ relaxation and $\{^1\text{H}\}$ - ^{15}N heteronuclear nuclear Overhauser effect (NOE) under two different magnetic fields (700 MHz and 800 MHz) (**Supplementary Fig. 7**). The average $^{15}\text{N}T_1$ relaxation times of the 15-mer were 538 and 794 ms at 700 and 800 MHz, respectively, whereas the average $^{15}\text{N}T_1$ relaxation times of the monomer were 416 and 420 ms at 700 and 800 MHz, respectively (Supplementary Fig. 7A and 7D). Furthermore, the average $^{15}\text{N}T_2$ relaxation times of the 15-mer at 700 and 800 MHz were 183 and 134 ms, respectively, which were shorter than the $^{15}\text{N}T_2$ relaxation times of the monomer, which were 189 and 169 ms at 700 and 800 MHz, respectively. The shorter $^{15}\text{N}T_1$ and longer of $^{15}\text{N}T_2$ relaxation times of the monomer suggested that the monomer had a shorter rotational correlation time and faster dynamics than the 15-mer³⁷.

The local dynamics (individual H-N bond vector) of the monomer and 15-mer were probed with $\{^1\text{H}\}$ - ^{15}N heteronuclear NOE experiments using 2 magnetic fields (700 and 800 MHz) (Supplementary Fig. 7C and 7F). Our results demonstrated that the 15-mer had higher NOE values at both 700 and 800 MHz NMR compared with the monomer, suggesting that the 15-mer was less flexible relative to the monomer. Furthermore, our dynamics data revealed higher NOE values of the 15-mer in the glycine-rich region than in the polyalanine region (**Fig. 3A**).”

Supplementary Figure 7: Backbone $^{15}\text{NT}_1$, $^{15}\text{NT}_2$ relaxation and $\{^1\text{H}\}$ - ^{15}N heteronuclear NOE of the 15-mer and monomer at pH 7, 10°C and two different magnetic fields: 700 MHz and 800 MHz. (A). Backbone $^{15}\text{NT}_1$ relaxation of the 15-mer at pH 7, 10°C, 700 MHz (red) and 800 MHz (black). (B). Backbone $^{15}\text{NT}_2$ relaxation of the 15-mer at pH 7, 10°C, 700 MHz (red) and 800 MHz (black) (C). $\{^1\text{H}\}$ - ^{15}N heteronuclear NOE of 15-mer at pH 7, 10°C and 700 MHz (red) and 800 MHz (black). (D) Backbone $^{15}\text{NT}_1$ relaxation of the monomer at pH 7, 10°C, 700 MHz (red) and 800 MHz (black). (E) Backbone $^{15}\text{NT}_2$ relaxation of the monomer at pH 7, 10°C, 700 MHz (red) and 800 MHz (black). (F) $\{^1\text{H}\}$ - ^{15}N heteronuclear NOE of the monomer at pH 7, 10°C and 700 MHz (red) and 800 MHz (black).

In addition, we also have provided additional experimental details for measuring $^{15}\text{NT}_1$, $^{15}\text{NT}_2$ relaxation and $\{^1\text{H}\}$ - ^{15}N heteronuclear NOE of the 15-mer and monomer at 700 MHz and 800 MHz and formulas to calculate the spectral densities in the Methods section (page 34-36, line 774-835).

Comment 5

What is the error in the NOE measurement presented in Figure 3? One can view that result as an average NOE value of $\sim 0.2 \pm 0.1$. And thus, they are essentially the same.

Response

According to the reviewer's comment, the NOE measurement errors were included in the revised manuscript (**Fig. 3A**). This NOE pattern for the repetitive domain was consistent even when the $\{^1\text{H}\}$ - ^{15}N heteronuclear NOE measurements were performed at different temperatures (Supplementary Fig.13C) and with different magnetic fields (Supplementary Fig 7C). The errors were small and determined based on the background noise level using the following formula (Ref 72: Farrow Biochemistry, 1994):

$$\sigma\text{NOE}/\text{NOE} = ((\sigma I_{\text{sat}}/I_{\text{sat}})^2 + (\sigma I_{\text{unsat}}/I_{\text{unsat}})^2)^{1/2}$$

We have added a description of the formula in the Methods section as follows:

Page 35, line 801-813

3. $\{^1\text{H}\}$ - ^{15}N heteronuclear NOE

“ $\{^1\text{H}\}$ - ^{15}N steady-state NOE values were obtained by measuring two spectra: an initial spectrum recorded without the initial proton saturation and a second spectrum recorded with the initial proton saturation (3 s). $\{^1\text{H}\}$ - ^{15}N steady-state NOE values of the monomer and 15-mer were determined at 700 and 800 MHz. The steady-state NOE values were determined from the ratios of the average intensities of the peaks with and without proton saturation, and the standard deviation of the NOE values was determined from the background noise level using the following formula⁷²:

$$\sigma\text{NOE}/\text{NOE} = ((\sigma I_{\text{sat}}/I_{\text{sat}})^2 + (\sigma I_{\text{unsat}}/I_{\text{unsat}})^2)^{1/2} \quad (3)$$

where I_{sat} and I_{unsat} are the measured intensities of the resonance in the presence and absence of proton saturation, respectively. The noise of the background regions of the spectra, which were recorded with the initial proton saturation and without initial proton saturation, is shown as σI_{sat} and σI_{unsat} , respectively.”

Error bars have been added to Figure 3A as follows:

Figure 3. Backbone dynamics and distribution of the secondary structure population of repetitive domains. (A) $\{^1\text{H}\}$ - ^{15}N heteronuclear NOE of the 15-mer.

Comment 6

Either way a slightly positive NOE at 700 MHz still indicates rapid ns backbone dynamics and a high degree of flexibility for all the residues, which is again in line with the random coil structure.

Response

The NOE values of the 15-mer and monomer were consistently different, even when they were measured using different magnetic fields (**Supplementary Fig. 7C and 7F**). The monomer always demonstrated more flexibility than the 15-mer. Furthermore, additional evidence was also provided by the $^{15}\text{NT}_1$ relaxation and $^{15}\text{NT}_2$ relaxation data for the 15-mer and monomer using two different magnetic fields. The shorter $^{15}\text{NT}_1$ relaxation times of the monomer compared with those of the 15-mer suggest that the monomer has a shorter rotational correlation time and faster dynamics than the 15-mer. In addition, the average $^{15}\text{NT}_2$ relaxation times of the 15-mer at 700 and 800 MHz were shorter compared with the $^{15}\text{NT}_2$ relaxation times of the monomer, suggesting that the 15-mer had a longer correlation time and slower motion than the monomer. Importantly, the spectral density at 0 MHz, which reflects the rotational correlation time of the molecule, indicated that the 15-mer had a longer rotational correlation time ($J(0) \sim 5.5$ ns) than the monomer ($J(0) \sim 2.2$ ns), suggesting that the 15-mer was not a fully random coil structure.

In addition to NMR spectroscopy data, we also measured the CD and VCD spectroscopy of the 15-mer and monomer. The data confirmed that the 15-mer contained a larger PPII helix population than the monomer, whereas the monomer predominantly contained random coil population. We also investigated the temperature effect on the conformation and dynamics of the repetitive domain by NMR and CD,

which showed that the PPII helix population was still present in the 15-mer at higher temperatures. Our CD data for the 15-mer (Supplementary Fig 13) demonstrated the presence of the PPII helix population at 30°C. The VCD data for the 15-mer at 25°C also showed the presence of the PPII helix population, whereas the monomer indicated the presence of the random coil population (Fig 1C). Based on these CD and VCD results, the 15-mer contained the PPII helix population while the monomer predominantly contained the random coil population. These additional experiments and discussion have been included in the revised manuscript as follows.

Page 10-11, line 221-259

“Backbone dynamics of the repetitive domains

The backbone dynamics of the monomer and 15-mer were probed with the ^{15}N T₁, ^{15}N T₂ relaxation and $\{^1\text{H}\}$ - ^{15}N heteronuclear nuclear Overhauser effect (NOE) under two different magnetic fields (700 MHz and 800 MHz) (**Supplementary Fig. 7**). The average ^{15}N T₁ relaxation times of the 15-mer were 538 and 794 ms at 700 and 800 MHz, respectively, whereas the average ^{15}N T₁ relaxation times of the monomer were 416 and 420 ms at 700 and 800 MHz, respectively (Supplementary Fig. 7A and 7D). Furthermore, the average ^{15}N T₂ relaxation times of the 15-mer at 700 and 800 MHz were 183 and 134 ms, respectively, which were shorter than the ^{15}N T₂ relaxation times of the monomer, which were 189 and 169 ms at 700 and 800 MHz, respectively. The shorter ^{15}N T₁ and longer of ^{15}N T₂ relaxation times of the monomer suggested that the monomer had a shorter rotational correlation time and faster dynamics than the 15-mer³⁷.

The local dynamics (individual H-N bond vector) of the monomer and 15-mer were probed with $\{^1\text{H}\}$ - ^{15}N heteronuclear NOE experiments using 2 magnetic fields (700 and 800 MHz) (Supplementary Fig. 7C and 7F). Our results demonstrated that the 15-mer had higher NOE values at both 700 and 800 MHz NMR compared with the monomer, suggesting that the 15-mer was less flexible relative to the monomer. Furthermore, our dynamics data revealed higher NOE values of the 15-mer in the glycine-rich region than in the polyalanine region (**Fig. 3A**).

The ^{15}N T₁, ^{15}N T₂ $\{^1\text{H}\}$ - ^{15}N heteronuclear NOE data for the monomer and 15-mer using 2 different magnetic fields, 700 and 800 MHz, were then translated into spectral densities, which revealed the molecular motion at certain angular frequency. Those spectral densities of the 15-mer and monomer were calculated at $J(0)$, $J(\omega\text{N})$ and $J(0.87\ \omega\text{H})$, which corresponded to 0, 70, 80, 609 and 696 MHz (**Supplementary Fig. 8**). The spectral densities $J(609\ \text{MHz})$ and $J(696\ \text{MHz})$ of the monomer were longer (~0.03 -0.04 ns/rad) than those of the 15-mer (~0.01-0.02 ns/rad) (Supplementary Fig. 8A). Furthermore, the spectral densities $J(70\ \text{MHz})$ and $J(80\ \text{MHz})$ of the monomer were not significantly different, whereas the spectral density $J(80\ \text{MHz})$ of the 15-mer decreased in comparison to its spectral density at $J(70\ \text{MHz})$ (Supplementary Fig. 8B). Those results also indicated that the monomer underwent faster motion than the 15-mer^{38,39}.

The most striking results were obtained for the spectral densities of the 15-mer and monomer at a frequency of 0 MHz ($J(0\ \text{MHz})$) (Supplementary Fig. 8C) because this spectral density reflects the rotational correlation times (τ_c) of those molecules. The $J(0)$ values of 15-mer (~5.5 ns) suggested that the rotational correlation time

of the 15-mer was longer than the rotational correlation time of the monomer, which presented a $J(0)$ of approximately 2.2 ns. This finding also indicated that the 15-mer underwent slower motion compared with the monomer. Taken together, all the dynamics data indicated that the 15-mer did not behave as a fully random coil protein because it rotated more slowly than intrinsically disordered proteins, which normally have a very short spectral density $J(0)$ (0.5-2.2 ns)³⁹⁻⁴¹,

Page 16, line 347-349

“The $\{^1\text{H}\}$ - ^{15}N heteronuclear NOE data for the repetitive domain consistently showed that the glycine-rich region had more limited flexibility compared with the polyalanine-region, even at a higher temperature (15°C) (**Supplementary Fig. 13C**).”

Page 6, line 136-152

“The structural propensities of the repetitive domain of spider silk at low concentration (10–20 μM) were determined via CD spectroscopy. The monomer spectra had a minimum at approximately 200 nm, which is typical of an intrinsically disordered protein^{29,30}. In contrast, the spectra of the dimer, trimer, hexamer and 15-mer demonstrated minima at approximately 196 nm; in particular, the CD spectra of the hexamer and 15-mer displayed small but detectable maxima at approximately 215 nm (**Fig. 1B**), which is consistent with the PPII helix conformation^{29,30}. Importantly, similar minima were detected in the CD spectra of multiple repetitive domains (dimer, trimer, hexamer and 15-mer) as the spectra of native spider silk proteins from *Nephila edulis* in fresh conditions (incubation time $t=0$)³¹.

The presence of the PPII helix population of the repetitive domains were evaluated in D_2O at a high concentration (8 mM) via VCD spectroscopy in the amide I region corresponding to the carbonyl stretch. The VCD spectrum of the 15-mer at pH 7 and 25°C in the amide I region demonstrated a maximum at 1662 cm^{-1} and an observable minimum at 1636 cm^{-1} (**Fig. 1C**), suggesting the presence of the PPII helix^{23,32}. In contrast, the delta absorbance of the VCD spectrum of the monomer decreased significantly, indicating that the monomer predominantly contains the unordered random coil population rather than the PPII helix population³³.”

Comment 7

The strongest NMR evidence for the PPII structure are the 3-bond J-coupling measurements. However, it is not clear if these measurements were made for the other amino acids in the glycine-rich region. How do they compare to the PPII structure. The PPII structure exhibits similar chemical shifts to random coil and similar 3-bond J-couplings for the most part. Within the error of the measurement the random coil and PPII or similar. Comparisons between the PPII helix and random coil three bond J-couplings for the other non-Gly amino acids would go a long way in removing this ambiguity (e.g. Gln, Tyr, Leu with value reported separately).

Response

Based on a previous study (Ref 44: Lam & Hsu, Biopolymer, 2003), ^3J HNHA coupling

constants measured in model PPII helix peptides were between 6.3 Hz and 6.8 Hz, with an average value of 6.5 Hz. The amino acids found in the model PPII helix peptide were Tyr, Gly, Arg, Lys and Gln. In that study, no data were obtained for the ^3J HNHA coupling constant of Gly and Tyr. Our study demonstrated that the average ^3J HNHA coupling constant of the glycine-rich region of the 15-mer was 6.5 Hz (standard deviation \pm 0.5 Hz), which is closer to the average ^3J HNHA coupling constant for a peptide containing PPII helix (6.5 Hz) (Ref 37). For a more specific comparison, we found that the average ^3J HNHA coupling constant of Gln of the 15-mer at 10°C was approximately 6.6 Hz (standard deviation \pm 0.2 Hz), which is similar to the previously reported ^3J HNHA coupling constant of the model PPII helix peptide at 10°C (6.8 Hz) (Ref 44). Unfortunately, the ^3J HNHA coupling constant of Gly, Leu and Tyr of the PPII helix peptide has not been previously reported. Therefore, we cannot compare the ^3J HNHA coupling constants of Gly, Leu and Tyr of the 15-mer and those of the model PPII helix peptide. Furthermore, in this study, we could not determine the ^3J HNHA coupling constant of Arg of the 15-mer due to overlapping signals.

In contrast, the ^3J HNHA coupling constant of the monomer in the glycine-rich region showed more variation (6.4, with standard deviation \pm 0.7 Hz), and in particular, the ^3J HNHA coupling constant of Gln of the monomer was 7.3, 6.9 and 5.9 Hz for Gln-8, Gln-21 and Gln-30, respectively, demonstrating that the actual ^3J HNHA coupling constants of Gln of the monomer were closer to the coupling constant value of Gln normally found in the random coil peptide (7.1 Hz) (Ref 44)

To clarify the ^3J HNHA coupling constants, we have added an additional explanation for this point in the Results section as follows:

Page 12, line 273-281

“For a more specific comparison, we found that the average ^3J HNHA coupling constant of Gln of the 15-mer at 10°C was approximately 6.6 \pm 0.2 Hz, which is similar to the previously reported ^3J HNHA coupling constant of the model PPII helix peptide at 10°C (6.8 Hz)⁴⁴. In comparison, the average ^3J HNHA coupling constant of the glycine-rich region of the monomer was 6.4 \pm 0.7 Hz, and the ^3J HNHA coupling constant of Gln of the monomer was 7.3, 6.9 and 5.9 Hz for Gln-8, Gln-21 and Gln-30, respectively, demonstrating that the actual ^3J HNHA coupling constants of Gln of the monomer were closer to the coupling constant value of Gln that is normally found in a random coil peptide (7.1 Hz)⁴⁴.”

Comment 8

This reviewer is not convinced that these small effects in the NMR are not due to entanglement for the higher MW recombinant silk proteins. At the concentrations studied, entanglement will have more of an impact on the higher MW protein.

Response

Regarding the comment from the reviewer about the contribution of greater entanglement of higher molecular weight polymers, we re-considered that entanglement might have an impact on the higher molecular weight of the repetitive domain at the concentration studied herein. Based on our NMR data, we have described the following points in the text.

1. We found that the differences in chemical shifts, J-coupling and dynamics of repetitive domain in different repeat units were caused by the presence of the PPII helix population, particularly in the glycine-rich region. This finding was also consistent with the CD data, which demonstrated the presence of the PPII helix population in the repetitive domain consisting of repeat units ≥ 2 . Our data also demonstrated that when the number of repetitive domains increased, the helical propensity of the GLGSQGTS region also increased, suggesting that the PPII helix population was stabilized by the larger number of repetitive domains (**Supplementary Fig. 3**). The PPII helix stabilization in the repetitive domain might be related to limited flexibility, particularly in the GLGSQGTS region. Furthermore, this PPII helix stabilization in larger numbers of repetitive domains enables spidroin to readily form intramolecular interaction and explains why stronger spider silk fibers can be achieved when the silk protein contains high numbers of repetitive domains (ref 20).
2. We found no change in the chemical shift and signal-to-noise ratio of the 15-mer at different concentrations.

According to those results, we did not clearly observe the entanglement effect under our conditions. The presence of the PPII helix and the limited flexibility in the glycine-rich region via intramolecular interactions explains why the soluble repetitive domain did not show entanglement effects. In the presence of mechanical and/or chemical stimuli (e.g., shear forces, kosmotropic ions), the entanglement effect of the repetitive domain might be affected at higher concentrations.

To clarify this point related to the concentration effect, we have added an additional explanation as follows.

Page 15-16, line 324-338

“Concentration effect on the repetitive domain

The intermolecular interactions of the repetitive domain were also evaluated in this study by monitoring the backbone chemical shift (N^H , H^N , $C\alpha$, $H\alpha$) of the 15-mer as a function of the concentration. Based on our NMR data, we found no change in the backbone chemical shifts of the repetitive domain at different concentrations (**Supplementary Fig. 12A, 12B and 12C**). This finding indicated that there was no change in the population of soluble repetitive domains, as indicated by the CD and VCD spectra (Fig. 1B and 1C). Furthermore, the ratio between the signal-to-noise (S/N) values of the signals from the 2D 1H - ^{15}N HSQC spectrum of the 15-mer at 1 mM (4 scans) and 0.5 mM (16 scans) was approximately one and uniform for the entire sequence. This result suggested that observable changes did not occur in the interaction or dynamics of the repetitive domain at higher concentrations (**Supplementary Fig. 12C**). This finding also suggested that no entanglement effect was observed for the repetitive domain under this condition. In the presence of mechanical and/or chemical stimuli (e.g., shear forces, kosmotropic ions, etc.), the entanglement effect of the repetitive domain might be affected at higher concentrations.”

Comment 9:

The authors discuss no change in chemical shifts at the different concentrations but, what about the dynamics?

Response

In this study, we calculated the signal-to-noise ratio of the signals from the 2D ^1H - ^{15}N HSQC spectra of the 15-mer with two-fold serial dilution (16 scans) and the 15-mer without dilution (4 scans). We found that the signal-to-noise ratio of the 15-mer with and without dilution was approximately one and was uniform over the entire sequence (**Supplementary Fig. 12A, B and D**). Therefore, under this condition, we consider that there were no changes in dynamics or interactions. This phenomenon can be explained as follows: 1 mM of molecules (concentration of the 15-mer in this study) is equal to 1 mol/m³ or approximately 10^{24} molecules per m³ or 1 molecule per 10^6 Å³. Assuming that 1 molecule is present in a cube, then the cube length required for 1 molecule from a 1 mM concentration is approximately 100 Å. In contrast, if the distance between amino acids is approximately 3.5 Å for the linear peptide, then the total length required for 1 molecule of the 15-mer (558 amino acids) in the extended form is 1953Å. Based on the present analysis, the soluble repetitive domain was clearly not in the extended form. The presence of the PPII helix and the limited flexibility in the glycine-rich region explains why the soluble repetitive domain was not in the extended form and no entanglement effect was observed in this condition. In the presence of mechanical and/or chemical stimuli (e.g., shear forces, kosmotropic ions), the entanglement effect of the repetitive domain might be affected at higher concentrations, which is beyond the scope of this study. To clarify the concentration effect on the repetitive domain, we have added an additional explanation in the Results section as follows.

Page 15, line 330-338 (a part of the response to comment 8)

“Furthermore, the ratio between the signal-to-noise (S/N) values of the signals from the 2D ^1H - ^{15}N HSQC spectrum of the 15-mer at 1 mM (4 scans) and 0.5 mM (16 scans) was approximately one and uniform for the entire sequence. This result suggested that observable changes did not occur in the interaction or dynamics of the repetitive domain at higher concentrations (**Supplementary Fig. 12C**). This finding also suggested that no entanglement effect was observed for the repetitive domain under this condition. In the presence of mechanical and/or chemical stimuli (e.g., shear forces, kosmotropic ions, etc.), the entanglement effect of the repetitive domain might be affected at higher concentrations.”

Supplementary Figure 12: Concentration effect on the repetitive domain. (A) ^1H - ^{15}N HSQC spectra of the 15-mer without dilution (1 mM); number of scans=4. (B) For two-fold dilutions (0.5 mM), the number of scans=16. (C) Overlay of the $\text{C}\alpha$ - $\text{H}\alpha$ region of aliphatic ^1H - ^{13}C HSQC of the 15-mer at different concentrations. Green is without dilution (concentration=1 mM, number of scans=4), and red is with two-fold dilutions (concentration=0.5 mM, number of scans=16). (D) Signal-to noise (S/N) ratio values of ^1H - ^{15}N HSQC spectra of the 15-mer (concentration 1 mM, number of scans=4) and the 15-mer (concentration 0.5 mM, number of scans=16) as a function of the residue number. All spectra are processed in the same manner, and certain overlapping peaks are not included in this graph.

Comment 10

And the 3-bond J-couplings? No change in the chemical shifts is consistent with a random coil but, what about the other measurements?

Response

As shown in **Supplementary Fig. 12A, B and C**, no change was observed in the backbone chemical shifts ($\text{C}\alpha$, $\text{H}\alpha$, N^{H} , H^{N}) at different concentrations of the repetitive domain. Therefore, this finding suggested that no change occurred in the 3-bond J-

couplings constant. In addition to the ^1H - ^{15}N HSQC spectra of the repetitive domain at different concentrations, we have also added the overlay of the $\text{H}\alpha$ - $\text{C}\alpha$ region from the ^1H - ^{13}C HSQC spectra of the repetitive domain at different concentrations (**Supplementary Fig. 12C**). Because no changes were observed in the chemical shifts, this finding also indicated that there were no changes in the population of the prefibrillar form of the repetitive domain of spider silk proteins, which was also demonstrated by the CD and VCD spectra (Fig. 1B and 1C). An explanation for this point has been added to the manuscript as follows:

Page 15, line 327-330 (a part of the response to comment 8)

“Based on our NMR data, we found no change in the backbone chemical shifts of the repetitive domain at different concentrations (**Supplementary Fig. 12A, 12B and 12C**). This finding indicated that there was no change in the population of soluble repetitive domains, as indicated by the CD and VCD spectra (Fig. 1B and 1C).”

Comment 11

The authors claim that because the repetitive domain displays no pH effect (conversion to β -sheet at acidic pH) then, the C and N-termini must be needed for this pH dependent β -sheet formation. Even if this is the case, one would need to engineer the repetitive domain with the C/N-termini and show that it did convert to β -sheet when the pH was changed. This experiment has not been done. And thus, this discussion is completely speculative. There could be countless reasons for why the repetitive domain does not convert to β -sheet at acidic pH that have nothing to do with the presence of the termini or not.

Response

This experiment has already been done in prior research reported in the literature. Based on several previous studies, pH-dependent β -sheet formation of spider silk protein has been investigated using native and recombinant spider dragline silk protein, which includes NTDs, repetitive domains and CTDs (Ref 60: Xu *et al.* Biomacromolecules, 2015; Ref 61: Lecrec *et al.* Biopolymer, 2012, Ref 62: Lecrec *et al.* Biopolymer, 2013). According to the results from those studies, spider dragline silk protein forms a β -sheet at an acidic pH. However, the effect of pH only on the repetitive domain of spider dragline silk protein has not yet been clarified. Therefore, in this study, we focused on investigating the pH effect on the conformation of the repetitive domain. The current NMR results showed no chemical shift change at an acidic pH. This finding indicated that the conformation of soluble prefibrillar of spider silk proteins did not change in the spinning duct. Therefore, based on the results from the present and previous studies, we concluded that the CTD and NTD might be necessary for inducing β -sheet formation of spider silk during the early stage of the spinning process.

To clarify this point, we have added a further explanation in the Discussion section as follows:

Page 21, line 457-461

“Thus, based on the results of the present and previous studies, we concluded that

CTD and NTD might be necessary for inducing β -sheet formation in spider silk during the early stage of the spinning process. Furthermore, dehydration, shear forces, and changes in biochemical environment are also involved in β -sheet formation of spider silk proteins.”

Reviewer #2:

Comment 1

Manuscript “Conformation and dynamics studies of the soluble precursor repetitive domain of spider elucidate initiation step of β -sheet formation” by Oktaviani et al, represent interesting structural inside on spider silk core domain. In this work author used several artificial model peptides comprising different number of repetition units related to natural MaSp protein of *N. clavipes*. The peptides we studied using plethora of advanced solution NMR technique to elucidate the role of structural unit repetition on chain dynamic and population of secondary structures. The peptides were also studied at different pH, and concentration conditions, both factors playing important roles in the natural spinning process. Manuscripts presents novel and interesting findings on secondary structure distributions and flexibilities along the repetitive unit as well as on intra-chain interactions between the units. These findings allow proposal of a new structural model of the soluble spider silk domain, which exhibits pre-structured (prefibrillar) silk population, which might facilitate fast beta-sheet formation during the fiber spinning. They show also that the pH changes in the range, which occurs in the spinning duct, do not influence this pre-structuring.

These findings represent important contribution to understanding how the highly soluble proteins in the spider dope transform into insoluble structures in the tough fiber. Nevertheless, there are several major and minor issues, author should correct and comment, before publication in Nature Communication.

Response

We thank the reviewer for the positive comments and have addressed each comment below.

Comment 2

Introduction:

Minor comments :

Unclear statement: “Several studies indicated that the NTD and CTD display strong pH dependence and are essential....” pH dependence of what? Structures? Interactions?

Please clear this.

Response

We agree with the reviewer and have revised the statement.

Page 3, line 65-67

“Several studies have shown that the NTD and CTD display strong pH dependence in terms of conformation and are essential for controlling the pH-dependent assembly of silk proteins¹⁴⁻¹⁸.”

Comment 3

The placement of the statement: “A recent report also showed the importance of tyrosine-tyrosine interaction in repetitive domain of silkworm silk to promote β -sheet formation” in the section about CTD and NTD is confusion; it might be more suitable in the section with pH changes.

Response

We agree with the reviewer and have moved the statement to the section related to pH changes in the Introduction. Furthermore, we have also added a further explanation to the section related to pH changes as follows:

Page 4-5, line 96-104

“Although the NTD and CTD display pH-dependent conformations, the pH-gradient effect in the gland on the repetitive domain of spider silk proteins still remains unknown. The importance of the pH effect on the repetitive domain was highlighted based on a recent report, which demonstrated the importance of dityrosine formation in the repetitive domain of silkworm silk to promote β -sheet formation²⁶. The formation of dityrosine in the repetitive domain was catalyzed by the presence of peroxidase²⁶, in which the activity of peroxidase is optimal at a slightly acidic pH (pH 6.0-6.5)²⁷. Thus, investigations of the pH effect on the conformation of the repetitive domain might be useful for identifying the comprehensive mechanism of β -sheet formation of spider silk, which until now has remained poorly understood.”

Comment 4

Results

CD Spectroscopy (Figure 1)

(A). Poor quality of the spectra from several points of view: Wrong scale or unit of molar ellipticity on Y-axis. Typical minimum at 200 nm should be at -5000 to -10000 deg.cm²/dmol. The intensities here are actually 100 times lower, than expected. Unexpectedly high CD intensity of the dimer at 200 nm in comparison to monomer and trimer points to rather miscalculation of the peptide concentration. Please correct this.

Response

Thank you very much for pointing out our oversight. Previously, in **Figure 1B**, we omitted (x 1000) on the Y-axis. We apologize for that oversight and have revised **Figure 1B** (see below). We have also corrected the concentration of the dimer and CD spectrum of the 15-mer (we repeated the measurement to generate a new CD spectrum of the 15-mer at pH 7). We also added vibrational circular dichroism (VCD) spectra for the 15-mer and monomer in D₂O at pH 7 and 25°C.

Figure 1. Repetitive domains of spider silk except for the monomer display the PPII helix conformation. (A) Repetitive domain of spidroin (i.e., monomer, dimer, trimer, hexamer and 15-mer) used in this study. The repeat amino acid sequence is indicated. (B) CD spectra of the different repeat constructs. The monomer (cyan) exhibited a minimum at approximately 200 nm, whereas the dimer (blue), trimer (green), hexamer (red), and 15-mer (black) displayed minima at approximately 196 nm (indicated in shaded region). The hexamer and 15-mer demonstrated slight maxima at approximately 215 nm (indicated in the shaded region and insert). (C) VCD spectra of the 15-mer and monomer at 25°C and pH 7 in D₂O. The 15-mer showed a distinctive maximum at 1662 cm⁻¹ and minimum at 1636 cm⁻¹, which is typical of the PPII helix population^{23,32}. In contrast, the overall delta absorbance of the monomer was significantly lower than that of the 15-mer, suggesting a higher proportion of the random coil conformation

Comment 5

The spectra of the 15-mer are claimed to be similar to native protein in ref. 25, but neither the intensity nor the profile are similar. Authors omit any comments to the unusual peak at 230 nm, which is very untypical for such kind of proteins. I suspect some impurities in this sample. Please check this.

Response

We apologize for this confusion. According to the reviewer's comment concerning the unusual peak at 230 nm that might have originated from impurities in the sample or cuvette, we repeated the CD measurement for the 15-mer at pH 7 and 10°C. The results indicated that the unusual peak was not present in the new CD spectrum and thus could have been derived from impurities, as the reviewer suspected. The new CD spectrum has been added to Figure 1B. In addition, based on the recent definition of the PPII helix population from Ref 29 (Greenfield, Nature protocol, 2006), the new CD spectrum should contain the PPII helix population. To clarify this issue, a further explanation has been provided as follows.

Page 6, line 142-144

“Importantly, similar minima were detected in the CD spectra of multiple repetitive domains (dimer, trimer, hexamer and 15-mer) as the spectra of native spider silk proteins from *Nephila edulis* in fresh conditions (incubation time $t=0$)³¹.”

Comment 6

Authors should consider the re-evaluation of the CD spectra. Additionally, I found this CD section only supportive for the whole story, and it would be more suitable as supporting information

Response

Recently, CD spectra have been used to distinguish peptides, which have a pure random coil population or contain the PPII helix population (Ref 29: Greenfield, Nature protocol, 2006). In this manuscript, we showed that the monomer, which consists of only one repeat unit, has minima at approximately 200 nm, which is typical of a random coil peptide/protein. In contrast, the dimer, trimer, hexamer and 15-mer, which contain 2, 3, 6 and 15 repeat units, respectively, consistently showed minima at approximately 196 nm and a weak positive band at approximately 215 nm, which indicates the presence of the PPII helix population. In addition to the NMR data, we consider the CD spectra of the repetitive domain in different-length repeat units to be informative and supportive as a first indication of the presence of the PPII helix population in the repetitive domain. To clarify the presence of the PPII helix population, we have added a red shadow in Figure 1B to indicate the minima at 196 nm and weaker positive band at 215 nm, thus denoting the presence of the PPII helix population. In addition, to confirm the presence of the PPII helix population in the repetitive domain, we have also added VCD spectra for the 15-mer and monomer in D₂O at pH 7 and 25°C. These data demonstrated that the 15-mer contained the PPII helix population as shown by the presence of distinctive maxima at 1662 cm⁻¹ and observable minima at 1636 cm⁻¹, which is typical of the VCD spectrum containing the PPII helix population. In contrast, the delta absorbance of the VCD spectrum of the monomer decreased significantly, strongly indicating a larger population of the random coil. To clarify these findings related to the PPII helix population, we have added additional details as follows.

Page 6, line 134-152

“Structural propensity of the repetitive domains at low and high concentrations by CD and VCD spectroscopy

The structural propensities of the repetitive domain of spider silk at low concentration (10–20 μM) were determined via CD spectroscopy. The monomer spectra had a minimum at approximately 200 nm, which is typical of an intrinsically disordered protein^{29,30}. In contrast, the spectra of the dimer, trimer, hexamer and 15-mer demonstrated minima at approximately 196 nm; in particular, the CD spectra of the hexamer and 15-mer displayed small but detectable maxima at approximately 215 nm (**Fig. 1B**), which is consistent with the PPII helix conformation^{29,30}. Importantly, similar minima were detected in the CD spectra of multiple repetitive domains (dimer, trimer, hexamer and 15-mer) as the spectra of native spider silk proteins from *Nephila edulis* in fresh conditions (incubation time t=0)³¹.

The presence of the PPII helix population of the repetitive domains were evaluated in D₂O at a high concentration (8 mM) via VCD spectroscopy in the amide I region corresponding to the carbonyl stretch. The VCD spectrum of the 15-mer at pH 7 and 25°C in the amide I region demonstrated a maximum at 1662 cm⁻¹ and an observable minimum at 1636 cm⁻¹ (**Fig. 1C**), suggesting the presence of the PPII helix^{23,32}. In contrast, the delta absorbance of the VCD spectrum of the monomer decreased significantly, indicating that the monomer predominantly contains the unordered random coil population rather than the PPII helix population³³.”

Comment 7

Section on pH effects (Figure 4) I found experiments and comments to the titrations in Figure 4 are not relevant for the story, since these experiments follow the backbone and residue changes at pH 7 – 11 i.e. conditions not related to natural spinning dope or spinning process, which is indeed in focus of the paper. Authors should focus rather on the titration and changes below pH 7 as described in Figure 5.

Response

The importance of an alkaline pH in this study was based on two findings from previous studies as follows.

(1) Volumetric and spectrophotometric titration studies of native spidroin (*Nephila edulis*) demonstrated two apparent pK values: 6.74, which was contributed by the deprotonation of different groups, and 9.21, which was likely contributed by the deprotonation of Tyr side chain residues (Ref 63: Dicko et al., *Biochemistry*, 2004).

(2) Engineered ADF-3 (eADF-3) showed no phase separation at pH >8.5, which might be related to deprotonation of Tyr side chain residues (Ref 64: Exler, *Angew. Chem. Int. Ed.*, 2007).

We agree with the reviewer that the pH effect on the repetitive domain should be focused on pH titration below 7. Therefore, we have revised the Results section accordingly. Compared with the previous version, in which we showed the basic pH effect and acidic pH effect on repetitive domains, in this revised version, we focused on the acidic pH effect of the repetitive domain and then the basic pH effect on the repetitive domain. Furthermore, Figure 4 (versus the previous version) has been moved to the supporting information (Supplementary Fig. 10).

Comment 8

Discussion

Major comments:

Figure 6; Please revise the sentence in the legend: “The polyalanine region is blue, and the glycine-rich region is red.” To the sentence: “The polyalanine region is red, and the glycine-rich region is blue”. It is obvious and generally well described that the polyalanines undergo the structural transformation into beta sheets, whereas PPII helix either not change or turn to alpha-helix, as described also in this manuscript.

Response

We thank the reviewer for pointing out our oversight. We have corrected the legend of Figure 6 (now Figure 5) as follows.

“Figure 5. Proposed mechanism of β -sheet formation of spider dragline silk. (A) Spider major ampullate gland displays a strong pH gradient. (B) Two major structural populations of spider dragline silk proteins, random coil and PPII helix in the glycine-rich region, occur in soluble form. The presence of the PPII helix in the glycine-rich region is proposed as a soluble prefibrillar form of spider dragline silk. The polyalanine region is shown in red, and the glycine-rich region is shown in black. Schematic illustration of PPII helix interaction in the glycine-rich region is shown in the box. In response to dehydration, shearing forces, extensional flow and changes in the

biochemical environment, this prefibrillar form will generate spider dragline fibers through strong β -sheet interactions.”

Comment 9

More details could be included into Figure 6B: schematic formation of the PPII helices on the right site and schematic intra-chain interaction in the PPII regions

Response

According to the reviewer’s comment, we have revised Figure 6 (Figure 5 in the revised version) in terms of the schematic intra-chain interaction in the PPII regions, namely, we have added the schematic intra-chain interaction in the PPII regions (see below).

Figure 5. Proposed mechanism of β -sheet formation of spider dragline silk. (A) Spider major ampullate gland displays a strong pH gradient. (B) Two major structural populations of spider dragline silk proteins, random coil and PPII helix in the glycine-rich region, occur in soluble form. The presence of the PPII helix in the glycine-rich region is proposed as a soluble prefibrillar form of spider dragline silk. The polyalanine region is shown in red, and the glycine-rich region is shown in black. Schematic illustration of PPII helix interaction in the glycine-rich region is shown in the box. In response to dehydration, shearing forces, extensional flow and changes in the biochemical environment, this prefibrillar form will generate spider dragline fibers through strong β -sheet interactions.

Comment 10

Recent NMR studies on natural and recombinant spider silk protein and the influence of pH on the polypeptide chain structure also should be considered in the discussion:

Leclerc, J., Lefèvre, T., Pottier, F., Morency, L.-P., Lapointe-Verreault, C., Gagné, S. M. and Auger, M. (2012), Structure and pH-induced alterations of recombinant and natural spider silk proteins in solution. *Biopolymers*, 97: 337–346.

Leclerc, J., Lefèvre, T., Gauthier, M., Gagné, S. M. and Auger, M. (2013), Hydrodynamical properties of recombinant spider silk proteins: Effects of pH, salts and shear, and implications for the spinning process. *Biopolymers*, 99: 582–593.

Response

We have added the references in the Discussion (page 21, line 455-457, ref 61 and 62) as follows:

“However, in vitro studies of native and recombinant spider dragline silk composed of NTD, repetitive domains and CTD demonstrated β -sheet formation at an acidic pH⁶⁰⁻⁶².”

Comment 11

Supplementary Figure 2: B) and C) lettering is missing in the figure
Wrong description of Y-axes: Author claim in the legend to present 1H-15N-HSQC spectra, however, chemical shifts on the Y-axes are described as 13C. Please correct this.

Legend C): claimed aliased peak form 43.89 ppm, which however cannot be found in the spectra. Please correct this.

The scale of the X-axes in B and C are the same (8.8-7.8) but the dimension of the axes differ. Please adjust

Response

We apologize for these mistakes. According to the reviewer’s comment concerning the description of the y-axes and scale of **Supplementary Fig. 2**, we have corrected the description of the y-axes of **Supplementary Fig. 2B and 2C** and adjusted the scale in **Supplementary Fig. 2B and 2C**.

In relation to the legend of **Supplementary Fig. 2C**, an additional peak of the synthesized unlabeled monomer 121.895 ppm was previously considered as an aliased peak of the N terminus. However, as pointed out by the reviewer, this cannot be an aliased peak of the N terminus; thus, we have corrected **Supplementary Fig. 2** and its figure legend. The additional peak (8.414 ppm; 121. 895 ppm) is probably from the amide proton and amide nitrogen of L4.

A Repetitive domain with the the his-tag MHHHHHSSG LVPRGSGMKE TAAAKFERQH MDSPDLGTDD
 DDKAMAS[GR RGGLGGQGAGAAAAAGYGGLGSQG]

Monomer without Thr-Ser cloning site RGGLGGQGAGAAAAAGYGGLGSQG

Supplementary Figure 2: Different oligomers of the repetitive domain and the influence of the His-tag domain and Thr-Ser cloning site on the spectrum of the repetitive domain. (A) Amino acid sequences of the repetitive domain (black) with the His-tag domain (blue) (upper) and amino acid sequences of the monomer without the His-tag and Thr-Ser (red) cloning site (bottom). For the monomer, dimer, trimer, hexamer and 15-mer, n is 1, 2, 3, 6, and 15, respectively. (B) 2D ^1H - ^{15}N HSQC overlay of the monomer (red), dimer (blue), trimer (green), hexamer (purple) and 15-mer (yellow) at pH 7 and 10°C . As the number of repetitive domains increases, the His-tag signals decrease. (C) 2D ^1H - ^{15}N HSQC overlay of the monomer with (blue) and without (red) the His-tag at pH 7 and 10°C . The C-terminus is indicated. The additional peak (8.414 ppm; and 121.895 ppm) is likely caused by the amide proton and amide nitrogen signal of L4. The monomer spectra with and without the His-tag and Thr-Ser cloning site overlay very well, suggesting that the His-tag and Thr-Ser cloning site do not affect the conformation of the repetitive domain.

Reviewers' Comments:

Reviewer #1:

Remarks to the Author:

The authors have diligently addressed this reviewer's comments and provided justification for publication in Nature Communications. In this reviewer's opinion the manuscript can be published as is.

Reviewer #2:

Remarks to the Author:

Manuscript "Conformation and dynamics studies of the soluble precursor repetitive domain of spider elucidate initiation step of β -sheet formation" by Oktaviani et al:

The authors corrected all issues from the first round, and I recommend the manuscript for publication in Nat. Comm. The work introduce new insights into pre-structuring of spider silk proteins in spinning dopes which will impact current models of the natural silk spinning process.

Reviewer #3:

Remarks to the Author:

The work by Oktaviani et al. investigates the solution conformation and dynamics of the isolated repetitive domain of spider silk under a variety of conditions (concentration, pH, temperature). The authors study five different recombinant protein constructs with increasing number of repetitive sequences (monomer to 15-mer) and find that the repetitive domains, except for the monomer, display approx. 24% PPII helix and 65% random coil conformation. Based on the findings the authors propose an initial mechanism of β -sheet formation in spider dragline silk.

This manuscript represents a comprehensive evaluation using several biophysical techniques (CD, VCD and NMR), and makes a strong point about the conformation of the repetitive domain in solution. However, a major weakness is that the relevance of using the isolated repetitive domain in absence of the terminal domains is not adequately addressed. The authors have neglected recent findings that the repetitive domains in constructs containing the terminal domains are sequestered in micelles with the terminal domains in the shell and the repetitive regions shielded in the core (Lin et al. PNAS 2009, 106, 8906-8911; Andersson et al. Nat. Chem. Biol. 2017, 13, 262-264). The conformation and dynamics of the repetitive domains in such micelles might be different from that in solution as indicated by a recent paper (Otikovs et al. Angew. Chem. Int. Ed. 2017, 56, 12571-12575).

In general, the work is important, well written and similar articles on spider silk terminal domains have been published in Nat. Commun. (Schwarze et al. Nat. Commun. 2013, 4, 2815; Kronqvist et al. Nat. Commun. 2014, 5, 3254). Therefore, I do not think that this needs to go to a more specialized journal. I would recommend that the authors rewrite the manuscript to address the following issues:

- 1) The relevance of using the isolated repetitive domain in absence of the terminal domains should be discussed in connection with the findings that the repetitive domains in constructs with the terminal domains are sequestered in micelles (Lin et al. PNAS 2009, 106, 8906-8911; Andersson et al. Nat. Chem. Biol. 2017, 13, 262-264). Although the repetitive domains alone are able to form fibers, the mechanical properties are significantly diminished.
- 2) The fact that the repetitive domain has propensity for PPII helix conformation has been shown previously by Lefevre et al. (Biomacromolecules 2007, 8, 2342-2344). Therefore, it should be stated explicitly what is new in this study.
- 3) The authors write on p. 6 "the CD spectra of the hexamer and 15-mer displayed small but detectable maxima at approximately 215 nm (Fig. 1B), which is consistent with the PPII helix conformation". If the maximum (positive peak) at 215 nm is taken as proof for PPII helix

conformation (according to Lopes et al. *Protein Sci. Publ. Protein Soc.* 2014, 23, 1765-1772), then the dimer and trimer do not display PPII helix conformation, which is in contradiction with the caption (title) of Figure 1 and text in the discussion section.

4) The authors write on p. 7 that the concentrations of the NMR samples (1.5-2.2 mM) were comparable to the molar concentrations of spidroins in the major ampullate gland. It is not appropriate to compare molar concentrations, instead the % w/v or mg/mL concentrations should be compared, since it is much easier to achieve high molar concentrations using shorter repetitive regions.

5) For the comparison of backbone carbonyl chemical shifts (Supplementary figure 6) the authors have excluded the interface residues 1-2 and 32-33. It would be good to also avoid chain-end effects (for the monomer), which likely affect the chemical shifts of at least the last 3 residues. Therefore, the comparison should be made between dimer and 15-mer.

6) The increase in rotational correlation time for the 15-mer compared to the monomer is attributed to increased structural propensity. Since PPII helices show inter-chain hydrogen bonds (e.g. such as in collagen), the increased rotational correlation time could also be attributed to transient interactions between the repetitive sequences. This possibility should be discussed in the text.

7) A comparison of the conformation and dynamics of the repetitive domains free in solution (this paper) and in micellar form (Otkovs et al. *Angew. Chem. Int. Ed.* 2017, 56, 12571-12575) should be included in the discussion.

8) Figure 5B: what is the evidence that the poly-Ala regions are aligned in the soluble prefibrillar form? The data seem to indicate the opposite, i.e. that poly-Ala regions are more flexible than glycine rich regions. Please also discuss what are the types of interactions in the PPII helix (hydrogen bonds, hydrophobic interactions etc.).

Other minor issues:

1) Supplementary figure 1 caption: It is not mentioned what is Lane 1A (marker).

2) p. 6 line 143: "detected minima similar to those of the spectra of **native** spider silk". It seems that **regenerated** spider silk was analyzed in the referenced article.

3) Figure 1: please note in the caption at what temperature were the CD spectra recorded.

4) Figure 1 caption: "The monomer (cyan) exhibited a minimum at approximately 200 nm, whereas the dimer (blue), trimer (green), hexamer (red), and 15-mer (black) displayed minima at approximately 196 nm (indicated in shaded region)." According to the legend in the figure, the monomer is spectrum is black, hexamer cyan and 15-mer is red.

5) Supplementary figure 2: There is no "n" in panel A after the brackets.

6) Supplementary figure 2: It makes no sense to overlay all 5 spectra if the ones below cannot be seen. I suggest overlying only 2-3 spectra or showing them next to each other.

7) p. 9 line 196: "indicated by a low structural propensity value (<0.5)", <0.5 should be changed to the actual maximum value (<0.15 or <0.2).

8) p. 15 line 237: please indicate the concentration range.

Manuscript ID: NCOMMS-16-28855A-Z

Title: Conformation and dynamics of soluble repetitive domain elucidates the initial β -sheet formation of spider silk (Revised title)

Corresponding author: Keiji Numata, RIKEN Center for Sustainable Resource Science, Japan

RESPONSES TO REVIEWER COMMENTS

Reviewer #3:

Comment 1

In general, the work is important, well written and similar articles on spider silk terminal domains have been published in Nat. Commun (Schwarze et al. Nat. Commun. 2013, 4, 2815; Kronqvist et al. Nat. Commun. 2014, 5, 3254). Therefore, I do not think that this needs to go to a more specialized journal. I would recommend that the authors rewrite the manuscript to address the following issues:

Response: We thank the reviewer for his/her positive comments and careful evaluation of our manuscript. We have addressed each comment and revised the manuscript as described below.

Comment 2

The relevance of using the isolated repetitive domain in absence of the terminal domains should be discussed in connection with the findings that the repetitive domains in constructs with the terminal domains are sequestered in micelles (Lin et al. PNAS 2009, 106, 8906-8911; Andersson et al. Nat. Chem. Biol. 2017, 13, 262-264). Although the repetitive domains alone are able to form fibers, the mechanical properties are significantly diminished.

Response: As pointed out by the reviewer, we realized that in the absence of terminal domains, the mechanical properties of spider dragline silk are diminished (Ref 18: Heidebrecht et al. Adv. Mater. 2015 and Ref 45: Andersson et al. Nat. Chem. Biol. 2017). However, the previous studies demonstrated that individual domains in spider silk function independently from each other and there are no stable interactions between them (Ref 46: Otikovs, M *et al*, Angewandte chem Int Ed, 2017 and Ref 47: Lin, Z *et al*, PNAS, 2009). As shown by another previous study (Ref 17: Xia, X.-X. *et al* PNAS, 2010), repetitive domains are still able to form fibers in the absence of the terminal

domains and number of the repetitive domains are proportional to the strength of the recombinant silk fibers. Based on those studies, we therefore consider that our study on conformation and dynamics of the repetitive domain in the absence of terminal domains might be highly relevant to the conformations and dynamics of assembly of the repetitive domains in the presence of the terminal domains. To clarify this point, we have added an additional explanation in the Introduction section as follow:

Page 15-16, line 338-345

“In this study, we investigated the conformations and dynamics of the soluble repetitive domains of *N. clavipes* in the absence of the terminal domains. Although the mechanical properties of spider dragline silk are diminished in the absence of the terminal domains^{18,45}, previous studies reported that individual domains of spider silk function independently, and no stable interactions were observed between these domains^{46,47}. Therefore, we consider that our study on conformations and dynamics of repetitive domains in the absence of terminal domains might be highly relevant to the conformations and dynamics of the repetitive domains when considered in the presence of terminal domains.”

Comment 3

The fact that the repetitive domain has propensity for PPII helix conformation has been shown previously by Lefevre et al. (Biomacromolecules 2007, 8, 2342-2344). Therefore, it should be stated explicitly what is new in this study.

Response: The previous study had shown the propensity of PPII helix conformation in spider dragline silk using VCD spectroscopy (Ref 20: Levefre et al. Biomacromolecules 2007), which is consistent with our far-UV CD and VCD data. However, Levefre et al. provided only the overall conformation, while the conformation and dynamics of the repetitive domain as a function of each amino acid are still unknown. In the current study, using solution state NMR spectroscopy, we demonstrated the local conformation and dynamics of the repetitive domain as the function of the amino acid sequence. Our data clearly demonstrated that the soluble repetitive domain of spider silk protein consisted of two populations: random coil (~65%) and PPII helix (~24%). The PPII helix population was distributed over the glycine-rich region, whereas the random coil population was distributed over the polyalanine region. This finding is consistent with the dynamics data, which have consistently shown that the glycine-rich region has more limited flexibility compared with the polyalanine region. Thus, our study provides a

new and fundamental explanation for the initial mechanism of β -sheet formation. To clarify this point, we have added a further explanation in the Discussion section as follows:

Page 17-18, line 386-395

“The propensity of PPII helix conformation in native spider dragline silk had been reported using VCD spectroscopy²⁰, which is consistent with our far-UV CD and VCD data. This previous study using VCD spectroscopy provided the overall conformation, while the conformation and dynamics of the repetitive domain as a function of amino acid sequence remained unclear. In the present study, using solution state NMR spectroscopy, we demonstrated the local conformations and dynamics of the repetitive domain as the function of amino acid sequence. Our data clearly demonstrated that the PPII helix population was distributed over the glycine-rich region, whereas the random coil population was distributed over the polyalanine region. This finding is consistent with the dynamics data, namely, the glycine-rich region has more limited flexibility compared with the polyalanine region.”

Comment 4

The authors write on p. 6 "the CD spectra of the hexamer and 15-mer displayed small but detectable maxima at approximately 215 nm (Fig. 1B), which is consistent with the PPII helix conformation". If the maximum (positive peak) at 215 nm is taken as proof for PPII helix conformation (according to Lopes et al. Protein Sci. Publ. Protein Soc. 2014, 23, 1765-1772), then the dimer and trimer do not display PPII helix conformation, which is in contradiction with the caption (title) of Figure 1 and text in the discussion section.

Response: The monomer spectra had a minimum at approximately 200 nm, which is typical of an intrinsically disordered protein (Ref 26: Greenfield Nat. Protoc. 2006 and Ref 27: Lopes et al. Protein Sci. 2014). In contrast, the spectra of the dimer, trimer, hexamer and 15-mer demonstrated minima at approximately 196 nm (**Fig. 1B**). In addition, the CD spectra of the hexamer and 15-mer displayed small but detectable maxima at approximately 215 nm (**Fig. 1B**), which is consistent with the PPII helix conformation (Ref 26, 27). Based on this point, we have corrected the caption (title) of **Fig 1** (page 7) as follows:

“Figure 1. Longer repetitive domains (Hexamer and 15-mer) of spider silk display observable PPII helix conformation.”

In addition to CD spectra, our NMR data demonstrated that the helical propensity of the GLGSQGTS region also increased as the number of repetitive domains increased, suggesting that the PPII helix population was more stabilized by the greater number of repetitive domains (**Supplementary Fig. 3**). Furthermore, in **Supplementary Fig 5 and Supplementary Fig 6**, two carbonyl chemical shifts of GLGSQGTS region of the dimer, trimer and hexamer were detectable, corresponding to the carbonyl of GLGSQGTS residues in the middle region and the end terminal (disordered). As shown in **Supplementary Fig 5**, the peak intensities of the carbonyl chemical shifts of the GLGSQGTS region in the end terminal (disordered) decreased as the number of repetitive domain increased. In the case that the dimer and trimer are fully random coil, only backbone chemical shifts of residues $i-1$ and $i+1$ would be affected according to a previous report (Ref 32: Tamiola et al. J. Am. Chem. Soc. 2010). Since more distant residues of the dimer and trimer were affected (more than 2 residues from the interface of the repetitive domain), the dimer and trimer are not fully random coil. Our NMR data demonstrated that when the number of the repetitive domains was increased, the helical propensity of the GLGSQGTS region also increased, suggesting that the PPII helix population was stabilized by the greater number of repetitive domains (**Supplementary Fig. 3**). The CD spectra of the dimer and trimer did not display the positive peaks at 215 nm, which is a clear indication of PPII helix population. This might be because the PPII helix population is present but less stabilized in the dimer and trimer due to their short repeat lengths. To clarify this point, we have added more explanation in the Discussion section as follows:

Page 17, line 368-381

“Previously, solid-state NMR indicated that the conserved LG(G/S)QG motif in the glycine-rich region (**Supplementary Fig. 15**) adopts a turn conformation in MaSp1 fibers⁵⁴. Such findings are supported by our results, which showed that the structural propensity (**Supplementary Fig. 3**) and backbone carbonyl chemical shifts in the GLGSQGTS region differ from those of the monomer and other repetitive domains (**Supplementary Fig. 5 and 6**). These findings suggest that the chemical environment of this motif in multiple repetitive domains with at least 2 repeat units (dimer, trimer, hexamer and 15-mer), which are able to undergo intramolecular interactions, differs from that in the monomer. Our data also demonstrate that when the number of repetitive domains is increased, the helical propensity of the GLGSQGTS region also increases,

suggesting that the PPII helix population is stabilized by the greater number of the repetitive domains (**Supplementary Fig. 3**). In short repeat lengths, such as the dimer and trimer, the PPII helix population is present but less stable. Therefore, the CD spectra of the dimer and trimer did not display detectable positive maxima at 215 nm, which is a clear indication of a PPII helix population.”

Comment 5

The authors write on p. 7 that the concentrations of the NMR samples (1.5-2.2 mM) were comparable to the molar concentrations of spidroins in the major ampullate gland. It is not appropriate to compare molar concentrations, instead the % w/v or mg/mL concentrations should be compared, since it is much easier to achieve high molar concentrations using shorter repetitive regions.

Response: According to the comment, in addition to the mM concentrations of the repetitive domain, we also have added concentrations in mg/mL and have removed the sentence describing that the concentration of NMR samples was comparable to the molar concentrations of spidroins in the major ampullate gland as follows:

Page 7, line 168-170

“The concentrations of the (^{13}C , ^{15}N) repetitive domains of the spidroins used for the NMR measurements were approximately 1.5–2.2 mM (monomer: 11.7–17.0 g L⁻¹, dimer: 15.8–23.0 g L⁻¹, trimer: 19.6–28.8 g L⁻¹, hexamer: 31.4–46.0 g L⁻¹, 15-mer: 69.3–100 g L⁻¹).”

Comment 6

For the comparison of backbone carbonyl chemical shifts (Supplementary figure 6) the authors have excluded the interface residues 1-2 and 32-33. It would be good to also avoid chain-end effects (for the monomer), which likely affect the chemical shifts of at least the last 3 residues. Therefore, the comparison should be made between dimer and 15-mer

Response: According to the comment from reviewer, we have corrected **Supplementary Fig 6A** by excluding 3 residues: residues 1-3 and 30-32. In the case of the dimer, as shown in **Supplementary Fig. 5**, the backbone carbonyl peaks in GLGSQGTS consist of 2 peaks: one corresponds to the backbone carbonyl peak of GLGSQGTS in the middle region and the other one corresponds to the backbone carbonyl peak of GLGSQGTS in the end terminal. In contrast, the 15-mer showed only

the backbone carbonyl peaks of GLGSQGTS in the middle region, while the backbone carbonyl peaks of GLGSQGTS at the end terminal were not observable. To clarify this point, we have added an illustration (Supplementary Fig 5F), which describes the position of carbonyl chemical shifts of the repetitive domain (dimer) in the middle region and the end terminal.

Supplementary Figure 5: When another repetitive domain is added, the carbonyl chemical shift of the GLGSGQ (written in red, above) residues changes relative to that of the monomer. Carbonyl chemical shifts of T32, S33, G1 and R2 also change but are not shown in this figure. (A) Monomer, (B) dimer, (C) trimer, (D) hexamer, and (E) 15-mer. (F) illustration of the position of GLGSQG residues in the middle (indicated in red) and in the end terminal (indicated in green) of repetitive domain (dimer). Peaks corresponding to the His-tag domain are indicated in blue.

In addition to the chemical shift difference between backbone carbonyl chemical shift of 15-mer and monomer, we also added the chemical shift differences between the backbone carbonyl of 15-mer and dimer in the middle region and between the backbone carbonyl of the 15-mer and dimer at the end terminal (**Supplementary Fig 6B**), as follows:

Supplementary Figure 6: Carbonyl chemical shift differences between the 15-mer and monomer and between the 15-mer and dimer as function of residue number.

(A) Carbonyl chemical shift difference between the 15-mer and monomer as function of residue number. This plot does not include $^1\text{GRG}^2$ and $^{30}\text{QGT}^{32}$, which are located at the interface between the repetitive domain with the His-tag and interface between one repetitive domain and another repetitive domain, respectively. The positive deviation suggests that the helicity increases in the 15-mer relative to the monomer. The presence of extensive carbonyl chemical shift differences between the 15-mer and monomer, which are found beyond the interface of repetitive domain, strongly indicates that the 15-mer is not a completely random coil protein. (B) Carbonyl chemical shift differences between the 15-mer and dimer in the middle region (indicated in red square) and between the 15-mer and dimer in the end terminal region (indicated in black circle) as function of residue number. As shown in Supplementary Figure 5, the backbone carbonyl of the dimer in GLGSQG region consist of 2 peaks, which correspond to GLGSQG in the middle and end terminal of repetitive domain. The chemical shift difference between backbone carbonyl of the 15-mer and dimer in the middle is close to zero, except for Arg because this residue is in the interface between repetitive domain

and His-tag. In contrast, the chemical shift difference between backbone carbonyl of the 15-mer and dimer in the end terminal is similar to chemical shift difference between the 15-mer and monomer.

According to **Supplementary Fig 6B**, backbone carbonyl chemical shift of the 15-mer is similar to backbone chemical shift of the dimer in the middle region, except for Arg, because this residue is next to the His-tag. In contrast, the chemical shift difference between backbone carbonyl of the 15-mer and dimer in the end terminal is similar to the chemical shift difference between backbone carbonyl of 15-mer and monomer (**Supplementary Fig 6A**). As mentioned in the response to Comment 4, this finding indicates that the dimer is also not fully random coil. Furthermore, these results (**Supplementary Fig 5 and 6**) also suggest that the chemical environment of GLGSQG motif in multiple repetitive domains (dimer, trimer, hexamer and 15-mer), which are able to undergo intramolecular interactions, differs from the chemical environment of this motif in the monomer. To clarify these points, we have added more explanation as follows:

Page 9, line 206-216

“The carbonyl chemical shift differences between the 15-mer and monomer and between the 15-mer and dimer were plotted as a function of the residue number (**Supplementary Fig. 6A and 6B**). Since the backbone chemical shift of the intrinsically disordered protein is only influenced by direct neighbor residues ($i-1$ and $i+1$), chemical shift differences of more distant residues (more than 2 residues from the interface of repetitive domain) between the 15-mer and monomer indicated that the 15-mer was not a fully random coil protein (**Supplementary Fig 6A**)³⁵. In addition, the chemical shift difference between the 15-mer and dimer in the end terminal was similar to the chemical shift difference between the 15-mer and monomer, while the chemical shift difference between the 15-mer and dimer in the middle region was close to zero (**Supplementary Fig 6B**). This finding suggests that the dimer is not fully random coil, similar to 15-mer.”

Page 17, line 369-375

“Such findings are supported by our results, which showed that the structural propensity (**Supplementary Fig. 3**) and backbone carbonyl chemical shifts in the GLGSQGTS region differ from those of the monomer and other repetitive domains

(**Supplementary Fig. 5 and 6**). These findings suggest that the chemical environment of this motif in multiple repetitive domains with at least 2 repeat units (dimer, trimer, hexamer and 15-mer), which are able to undergo intramolecular interactions, differs from that in the monomer”

Comment 7

The increase in rotational correlation time for the 15-mer compared to the monomer is attributed to increased structural propensity. Since PPII helices show inter-chain hydrogen bonds (e.g. such as in collagen), the increased rotational correlation time could also be attributed to transient interactions between the repetitive sequences. This possibility should be discussed in the text.

Response: We agree with the comment and have added more explanation in the Results section as follow:

Page 10-11, line 241-244

“Additionally, higher rotational correlation time of the 15-mer might be also attributed to transient interaction between the repetitive domains since PPII helix conformation often shows intermolecular hydrogen bonds³⁶. (Ref 36: Bella & Berman J. Mol. Biol. 1996).”

Comment 8

A comparison of the conformation and dynamics of the repetitive domains free in solution (this paper) and in micellar form (Otikovs et al. Angew. Chem. Int. Ed. 2017, 56, 12571-12575) should be included in the discussion.

Response: As pointed out by the reviewer, conformation and dynamics of the recombinant repetitive domain of *N. clavipes* in this study are different from those of the recombinant repetitive domain of *Euprosthenoops australis* in the terminal domains (Ref 46: Otikovs et al. Angew. Chem. Int. Ed. 2017). These differences are possibly due to length of the polyalanine region. The long polyalanine region (14-15 alanine residues) of the repetitive domain of *E. australis* shows helical structure, while the glycine-rich region has random coil conformation. As shown in the current study, a shorter polyalanine region (5 alanine residues) in the *N. clavipes* repetitive domain leads to random coil conformation. Our data is in agreement with the previous study, which revealed that the conformation of polyalanine region (6-8 alanine residues) of

native *Lactrodectus hesperus* dragline silk is also random coil (Ref 56: Xu & Holland Biomacromolecules 2015). In contrast, the longer polyalanine region in *E. australis* repetitive domain leads this region to form helical structure (Ref 57: Hedhammar et al. Biochemistry 2008). In addition, this helical structure is also similar to the structure of a long polyalanine region (10-14 alanine residues) in *Samia cynthia ricini* (Ref 58: Suzuki et al. Macromolecules 2015). In the case of *E. australis* spidroins, hydrophobicity of the repetitive domain and hydrophilicity of the C-terminal domain cause to form micelle-like structure. Together, these studies suggest that the soluble form of shorter polyalanine region (4-8 alanine residues) of the repetitive domain in tends to form random coil, while longer polyalanine region (>10 alanine residues) tends to form helical structure.

Despite the difference in conformation and dynamics between the *N. clavipes* and *E. australis* repetitive domains, both of these repetitive domains show the combination of the glycine-rich and polyalanine regions in different dynamic behaviors. High flexibility of the polyalanine region and limited flexibility of the glycine-rich region of *N. clavipes* repetitive domain in the present study is contrary to the limited flexibility in the polyalanine region and high flexibility in the glycine-rich region of *E. australis* repetitive domain (Ref 46). Additionally, the recombinant repetitive domain of *E. australis* is easily aggregated, even at low concentration (1 mg/mL) (Ref 57), while the recombinant *N. clavipes* repetitive domain is soluble at relatively high concentration (100 mg/mL). This difference in solubility of the repetitive domains can be explained based on the dynamic behavior of the glycine-rich region. The limited flexibility of the glycine-rich region of the repetitive domain from *N. clavipes* seems to be essential for maintaining high solubility of spider silk protein by preventing premature aggregation, while the high flexibility of the glycine-rich region of *E. australis* repetitive domain leads this domain to be easily aggregated. We have added these explanations in Discussion section as follows:

Page 19-20, line 428-451

“Interestingly, conformation and dynamics of the recombinant repetitive domain of *N. clavipes* in this study are different from those of the recombinant repetitive domain of *Euprostenops australis* in the presence of terminal domains⁴⁶. These differences are possibly due to the length of the polyalanine region. The long polyalanine region (14-15 alanine residues) of the repetitive domain of *E. australis* shows helical structure, while the glycine-rich region has random coil conformation. As shown in the current

study, a shorter polyalanine region (5 alanine residues) in *N. clavipes* repetitive domain leads to random coil conformation. Our results are in agreement with previous study, which revealed that the conformation of polyalanine region (6-8 alanine residues) of native *Lactrodectus hesperus* dragline silk was random coil⁵⁶. In contrast, the longer polyalanine region in *E. australis* repetitive domain leads to the formation of helical structure⁵⁷. The helical structure of the polyalanine in *E. australis* is also similar to the helical structure of a long polyalanine region (10-14 alanine residues) in *Samia cynthia ricini*⁵⁸. In the case of *E. australis* spidroins, hydrophobicity of the repetitive domain and hydrophilicity of the C-terminal domain cause to form micelle-like structure⁴⁶. Together, these studies suggest that the soluble form of shorter polyalanine region (4-8 alanine residues) of the repetitive domain tends to form random coil, while longer polyalanine regions (>10 alanine residues) tend to form helical structure. Furthermore, the recombinant repetitive domain of *E. australis* is easily aggregated even at low concentration (1 g L⁻¹)⁵⁷, whereas the recombinant *N. clavipes* repetitive domain is soluble at relatively high concentration (100 g L⁻¹). The difference in solubility of the repetitive domains can be explained based on the dynamic behavior of the glycine-rich region. The limited flexibility of the glycine-rich region of the repetitive domain from *N. clavipes* seems to be essential for maintaining high solubility of spider silk protein by preventing premature aggregation. On the other hand, the high flexibility of the glycine-rich region of *E. australis* repetitive domain leads this domain to be easily aggregated⁵⁷.”

Comment 9

Figure 5B: what is the evidence that the poly-Ala regions are aligned in the soluble prefibrillar form? The data seem to indicate the opposite, i.e. that poly-Ala regions are more flexible than glycine rich regions. Please also discuss what are the types of interactions in the PPII helix (hydrogen bonds, hydrophobic interactions etc.)

Response: There is no evidence that the poly-Ala regions are aligned in the soluble prefibrillar form. Based on the reviewer’s comment, we have corrected Fig. 5B as follows:

Figure 5. Proposed mechanism of β -sheet formation of spider dragline silk. (A) Spider major ampullate gland displays a strong pH gradient. (B) Two major structural populations of spider dragline silk proteins, random coil and PPII helix in the glycine-rich region, occur in soluble form. The presence of the PPII helix in the glycine-rich region is proposed as a soluble prefibrillar form of spider dragline silk. The polyalanine region is shown in red, and the glycine-rich region is shown in black. Schematic illustration of PPII helix interaction in the glycine-rich region is shown in the box. In response to dehydration, shearing forces, extensional flow and changes in the biochemical environment, this prefibrillar form will generate spider dragline fibers through strong β -sheet interactions.

The PPII helix interaction in the glycine-rich region is possibly mediated by hydrogen bonds between alpha protons and oxygens from carbonyls. These are similar to the interactions (intra- and inter-chain) in the triple helix structure of collagen, namely, the hydrogen bonds between alpha proton of Gly and carbonyl oxygen of neighboring amino acid (Ref 36: Bella & Berman J. Mol. Biol. 1996). In the case of Gln, PPII helix interaction is possibly mediated by hydrogen bonds between Gln side chains and carbonyl oxygens of the neighboring residue, according to a previous study (Ref 55: Stapley & Creamer Protein Sci. 1999). Based on these discussion, we have added another explanation as follow:

Page 18, line 408-412

“The PPII helix interaction in the glycine-rich region is possibly mediated by hydrogen bonds between alpha proton and carbonyl oxygen, similar to the triple helix structure of collagen³⁶. In the case of Gln in the glycine-rich region, PPII helix interactions are possibly mediated by hydrogen bonds between Gln side chain and carbonyl oxygen of the neighboring residue⁵⁵.”

Comment 10

Supplementary figure 1 caption: It is not mentioned what is Lane 1A (marker).

Response: We thank the reviewer for pointing out the typo. We have corrected the caption of Supplementary Figure 1 as follow:

“Supplementary Figure 1: SDS-PAGE profile of the purified repetitive domains used in this study. (A) Marker, monomer (7.8 kDa), dimer (10.5 kDa) and trimer (13.1 kDa) are shown in Tris-tricine 10–20% SDS-PAGE Lanes 1A, 2A, 3A and 4A, respectively. (B) Hexamer is presented using 4–20% SDS-PAGE. Lanes 1B and 2B are the marker and hexamer (20.9 kDa), respectively. (C) Migration of the 15-mer (46.2 kDa) is shown using 4–15% SDS-PAGE. Lanes 1C and 2C are the marker and 15-mer, respectively.”

Comment 11

p. 6 line 143: "detected minima similar to those of the spectra of native spider silk". It seems that regenerated spider silk was analyzed in the referenced article.

Response: We thank the reviewer for pointing out our mistake. We have corrected that sentence with an appropriate reference as follows:

Page 6, line 139-141

“Consistently, the CD spectra of multiple repetitive domains (dimer, trimer, hexamer and 15-mer) detected minima similar to those of the spectra of regenerated spider silk proteins from *Nephila edulis* in fresh conditions (incubation time $t=0$)²⁸. (Ref 28: Shao et al. Macromolecules 2003).”

Comment 12

Figure 1: please note in the caption at what temperature were the CD spectra recorded.

Response: In Figure 1, CD spectra of the different repeat constructs were recorded at 10°C. We have added this information into the caption of Figure 1B as follows:

“(B) CD spectra of the different repeat constructs were recorded at 10°C.”

Comment 13

Figure 1 caption: "The monomer (cyan) exhibited a minimum at approximately 200 nm, whereas the dimer (blue), trimer (green), hexamer (red), and 15-mer (black) displayed minima at approximately 196 nm (indicated in shaded region)." According to the legend in the figure, the monomer is spectrum is black, hexamer cyan and 15-mer is red.

Response: We are very sorry for the typos. Based on the comment, we have corrected the caption of Figure 1 as follows:

“The monomer (black) exhibited a minimum at approximately 200 nm, whereas the dimer (blue), trimer (green), hexamer (cyan), and 15-mer (red) displayed minima at approximately 196 nm (indicated in shaded region).”

Comment 14 and 15

(5) Supplementary figure 2: There is no "n" in panel A after the brackets.

(6) Supplementary figure 2: It makes no sense to overlay all 5 spectra if the ones below cannot be seen. I suggest overlying only 2-3 spectra or showing them next to each other.

Response: Based on the comment, we have revised Supplementary Fig. 2 as follows:

Supplementary Figure 2: Different oligomers of the repetitive domain and the influence of the His-tag domain and Thr-Ser cloning site on the spectrum of the repetitive domain. (A) Amino acid sequences of the repetitive domain (black) with the His-tag domain (blue) (upper) and amino acid sequences of the monomer without the

His-tag and Thr-Ser (red) cloning site (bottom). For the monomer, dimer, trimer, hexamer and 15-mer, n is 1, 2, 3, 6, and 15, respectively. (B) 2D ^1H - ^{15}N HSQC overlay of the monomer (blue) and dimer (green). (C) 2D ^1H - ^{15}N HSQC overlay of the monomer (blue) and trimer (pink). (D) 2D ^1H - ^{15}N HSQC overlay of the monomer (blue) and hexamer (cyan) (E) 2D ^1H - ^{15}N HSQC overlay of the monomer (blue) and 15-mer (red). (F) 2D ^1H - ^{15}N HSQC overlay of the monomer with (blue) and without (maroon) the His-tag. All spectra were recorded at pH 7 and 10°C. The C-terminus is indicated. The additional peak (8.414 ppm; and 121.895 ppm) is likely caused by the amide proton and amide nitrogen signal of L4. The monomer spectra with and without the His-tag and Thr-Ser cloning site overlay very well, suggesting that the His-tag and Thr-Ser cloning site do not affect the conformation of the repetitive domain.

Comment 16

p. 9 line 196: "indicated by a low structural propensity value (<0.5)", <0.5 should be changed to the actual maximum value (<0.15 or <0.2).

Response: We have corrected the sentence as follows:

Page 8 line 189-192

“When the backbone chemical shifts of the repetitive domain were translated into structural propensity values using the neighbor-corrected structural propensity calculator (ncSPC)³¹, all repetitive domains were predominantly unfolded as indicated by a low structural propensity value (<0.15).”

Comment 17

p. 15 line 237: please indicate the concentration range.

Response: We have added the concentration range as follows:

Page 14, line 307-309

“Based on our NMR data, we found no change in the backbone chemical shifts of the repetitive domain at different concentrations (concentration range: 23.1 g L⁻¹ (0.5 mM) up to 46.2 g L⁻¹ (1 mM)) (Supplementary Fig. 12A, 12B and 12C).”

Reviewers' Comments:

Reviewer #3:

Remarks to the Author:

The revised manuscript by Oktaviani et al. convincingly addresses all of my concerns, and I therefore recommend acceptance. In my opinion the article absolutely merits publication in Nature Communications.